# Chunk-Guided Q-Learning

Gwanwoo Song [1]   Kwanyoung Park [2]   Youngwoon Lee [1]

## Abstract

In offline reinforcement learning (RL), single-step temporal-difference (TD) learning can suffer from bootstrapping error accumulation over long horizons. Action-chunked TD methods mitigate this by backing up over multiple steps, but can introduce suboptimality by restricting the policy class to open-loop action sequences. To resolve this trade-off, we present Chunk-Guided Q-Learning (CGQ), a single-step TD algorithm that guides a fine-grained single-step critic by regularizing it toward a chunk-based critic trained using temporally extended backups. This reduces compounding error while preserving fine-grained value propagation. We theoretically show that CGQ attains tighter critic optimality bounds than either single-step or action-chunked TD learning alone. Empirically, CGQ achieves strong performance on challenging long-horizon OG-Bench tasks, often outperforming both single-step and action-chunked methods. Project page: https://gwanwoosong.github.io/cgq

## 1. Introduction

Q-learning is a central approach in offline reinforcement learning (RL), yet it often struggles to learn accurate value functions on long-horizon tasks with sparse rewards (Park et al., 2025c; 2024; Park & Lee, 2025). A core issue lies in single-step temporal difference (TD) learning. Because the regression target depends on the critic's own value estimates, small errors compound across successive Bellman backups (Figure 1, left) (Sutton et al., 1998; Van Hasselt et al., 2018). In offline RL, this effect is amplified by limited data coverage, so target errors cannot be corrected by new interaction, and the accumulated error can overwhelm the learning signal and severely degrade performance as the

horizon grows (Park et al., 2025c).

Motivated by the observation that *horizon reduction* can mitigate bootstrapping error accumulation (Park et al., 2024; 2025c; Li et al., 2025b), we propose a simple yet effective way to leverage chunked backups by *guiding* a single-step critic with a chunk-based critic, gaining long-range stability while retaining fine-grained single-step compositionality.

Horizon reduction approaches reduce the effective backup depth using action chunking (Li et al., 2025b;a), $n$-step returns (Park et al., 2025c), or hierarchical RL (Park et al., 2024; 2025c), and have been shown effective in long-horizon, sparse-reward tasks. Notably, *action chunking* has been widely adopted in offline RL due to its simplicity and strong empirical performance (Li et al., 2025b;a; Park et al., 2025a; Kim et al., 2025).

However, this benefit comes at a cost in offline settings. In particular, $n$-step returns rely on multi-step off-policy rollouts, which can introduce bias when behavior and target policies differ (Hernandez-Garcia & Sutton, 2019), leading to suboptimal value estimates in long-horizon tasks (Park et al., 2025c). Action chunking exhibits a related but distinct limitation: it assumes open-loop execution over the entire chunk and therefore cannot properly value reactive decisions within the chunk (Li et al., 2025a), hindering fine-grained credit assignment and trajectory stitching (Figure 1, right).

These limitations raise a natural question:

> *"Can we leverage the stability benefits of horizon reduction without sacrificing the optimality and compositionality of single-step TD learning?"*

To address this challenge, we propose **Chunk-Guided Q-Learning (CGQ)**, which augments standard single-step TD learning with a regularizer that guides the critic toward an action-chunked critic trained via chunk-based TD updates. This combines the strengths of both views: the action-chunked critic provides a stable long-range learning signal, while the single-step critic preserves fine-grained value propagation and trajectory stitching.

Our contributions are threefold:

- We introduce CGQ, an offline RL method that effectively mitigates TD error accumulation by regularizing a single-step critic toward a chunk-based critic.

---

[1]Department of Artificial Intelligence, Yonsei University [2]UC Berkeley. Correspondence to: Youngwoon Lee <youngwoon@yonsei.ac.kr>.

*Proceedings of the 43rd International Conference on Machine Learning*, Seoul, South Korea. PMLR 306, 2026. Copyright 2026 by the author(s).

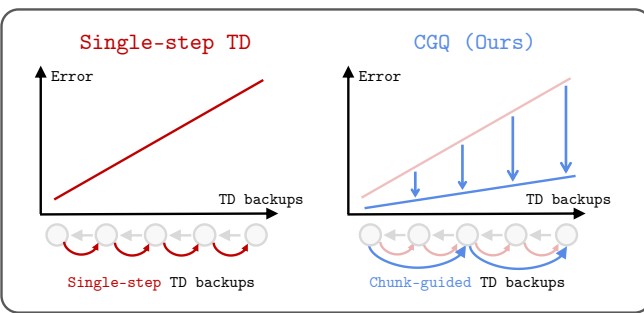 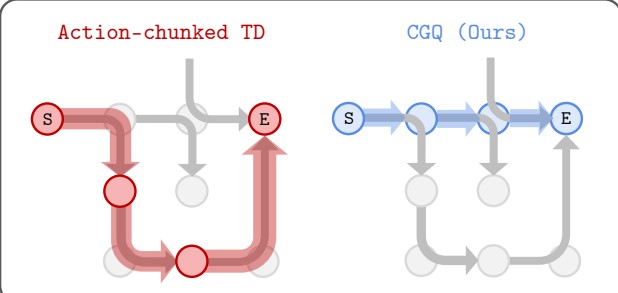

*Figure 1.* **Advantage of Chunk-Guided Q-Learning (CGQ) over single-step and action-chunked TD learning.** *(Left)* Single-step TD learning can suffer from compounding error because it bootstraps from its own value estimates; CGQ mitigates this by guiding the critic toward an action-chunked critic trained with temporally extended backups. *(Right)* Action-chunked TD learning can be suboptimal (in red arrows) because chunked TD backups assume open-loop action sequences, limiting fine-grained stitching across trajectories (gray arrows denote dataset trajectories). In contrast, CGQ can recover the optimal reactive policy (in blue arrows) by retaining single-step TD learning.

- We provide theoretical results showing CGQ yields tighter critic optimality bounds than either single-step or action-chunked TD learning alone, and give an intuitive explanation of when CGQ is beneficial.
- We demonstrate that CGQ often outperforms both action-chunked and single-step methods on long-horizon OGBench (Park et al., 2025b) tasks, surpassing prior action-chunking approaches.

## 2. Preliminaries

**Problem setting.** We consider a Markov Decision Process (MDP) defined as $\mathcal{M} = (\mathcal{S}, \mathcal{A}, p, r, \mu, \gamma)$, where $\mathcal{S}$ is the state space, $\mathcal{A}$ is the action space, $p(s' \mid s, a) : \mathcal{S} \times \mathcal{A} \to \Delta(\mathcal{S})$ is the transition dynamics, $r(s, a) : \mathcal{S} \times \mathcal{A} \to \mathbb{R}$ is the reward function, $\mu(s) \in \Delta(\mathcal{S})$ is the initial state distribution, and $\gamma \in (0, 1)$ is the discount factor.[1]

We study the offline RL setting, where the goal is to learn a policy $\pi$ that maximizes $\mathbb{E}\left[\sum_{t=0}^{\infty} \gamma^t r(s_t, a_t)\right]$ given a fixed dataset of transitions $\mathcal{D} = \{(s, a, r, s')\}$ collected by a behavior policy $\pi_{\mathcal{D}}$, without further environment interactions.

**TD learning.** We consider a parameterized policy $\pi_\theta(a \mid s)$ that will be optimized from the offline dataset and used for control. We learn a critic $Q_\phi(s, a)$ that estimates the discounted return, $\mathbb{E}_{\substack{s_0 = s, \\ a_0 = a}}[\sum_{t=0}^{\infty} \gamma^t r(s_t, a_t)]$ by minimizing the one-step temporal difference (TD) loss (Sutton, 1988):

$$\mathcal{L}^{\text{TD}}(\phi) = \mathbb{E}_{\substack{(s,a,r,s') \sim \mathcal{D}, \\ a' \sim \pi_\theta(\cdot|s')}} \left[ \left( Q_\phi(s, a) - r - \gamma Q_{\bar{\phi}}(s', a') \right)^2 \right], \tag{1}$$

where $Q_{\bar{\phi}}$ is a target network updated as an exponential moving average of $Q_\phi$ (Mnih et al., 2013).

**TD learning with action chunks.** Instead of training a critic that evaluates a single action $a_t$, we also con-

---

[1] $\Delta(\mathcal{X})$ denotes the set of probability distributions on space $\mathcal{X}$.

sider an action-chunked critic $Q_{\phi_c}(s_t, \mathbf{a}_t)$ over $h$-step action sequences $\mathbf{a}_t = (a_t, a_{t+1}, \cdots, a_{t+h-1})$. Let $\mathbf{r}_t = \sum_{i=0}^{h-1} \gamma^i r(s_{t+i}, a_{t+i})$ denote the $h$-step reward. We train $Q_{\phi_c}$ by minimizing the chunked version of Eq. (1):

$$\begin{aligned} \mathcal{L}_c^{\text{TD}}(\phi_c) = \mathbb{E}_{\substack{(s_t, \mathbf{a}_t, s_{t+h}) \sim \mathcal{D}, \\ \mathbf{a}_{t+h} \sim \pi_{\theta_c}(\cdot|s_{t+h})}} & \Big[ \big( Q_{\phi_c}(s_t, \mathbf{a}_t) \\ & - \mathbf{r}_t - \gamma^h Q_{\bar{\phi}_c}(s_{t+h}, \mathbf{a}_{t+h}) \big)^2 \Big], \end{aligned} \tag{2}$$

where $Q_{\bar{\phi}_c}$ is a target network for $Q_{\phi_c}$, and $\pi_{\theta_c}(\mathbf{a} \mid s)$ is an action-chunked policy. Since $Q_{\phi_c}(s_t, \mathbf{a}_t)$ evaluates open-loop action sequences, greedy chunk selection may fail to recover the optimal reactive policy in the original MDP.

**Flow Q-Learning (FQL).** A common challenge in offline RL is that policy improvement can select out-of-distribution actions, leading to unreliable value estimates. FQL (Park et al., 2025d) addresses this by learning an expressive flow-based behavior policy and constraining policy improvement toward the behavior distribution.

Concretely, FQL first fits a flow-based behavior policy $\pi_\theta(s, z) : \mathcal{S} \times \mathcal{A} \to \mathcal{A}$ by learning a velocity field $v_\theta$ with a flow matching loss (Lipman et al., 2023; Albergo & Vanden-Eijnden, 2023; Liu et al., 2023a)

$$\mathcal{L}^{\text{flow}}(\theta) = \mathbb{E}_{\substack{(s_t, a_t) \sim \mathcal{D}, \\ u \sim \text{U}([0,1]), \\ z \sim \mathcal{N}(0, I_\mathcal{A})}} \left[ \|v_\theta(u, s_t, a_z) - (a_t - z)\|_2^2 \right], \tag{3}$$

where $a_z = (1 - u)z + u a_t$. FQL then learns a constrained policy $\pi_\omega(s, z)$ by trading off Q-value maximization and distillation toward the flow-based behavior policy:

$$\begin{aligned} \mathcal{L}^{\text{FQL}}(\omega) = \mathbb{E}_{s \sim \mathcal{D}, z \sim \mathcal{N}(0, I_\mathcal{A})} \big[ & -Q_\phi(s, \pi_\omega(s, z)) \\ & + \alpha \|\pi_\theta(s, z) - \pi_\omega(s, z)\|_2^2 \big]. \end{aligned} \tag{4}$$

where $\alpha$ controls the strength of behavior regularization toward $\pi_\theta$. In this paper, we use FQL for single-step policy extraction.

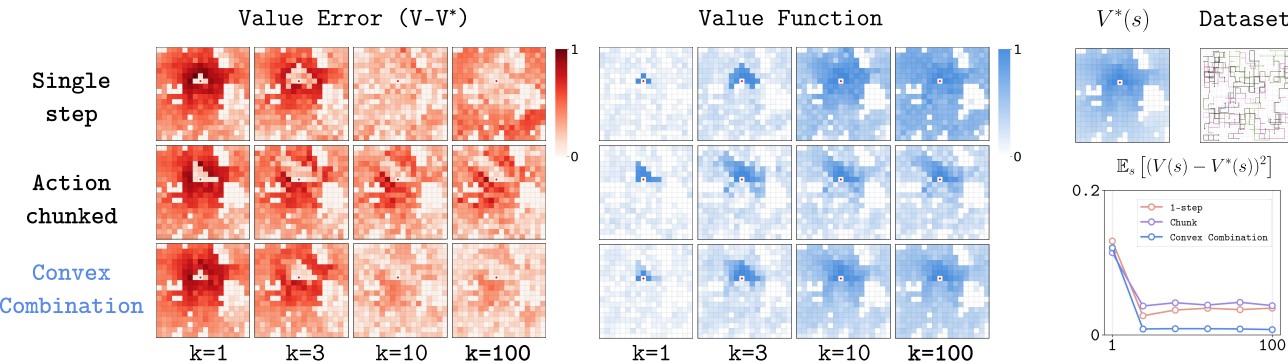

*Figure 2.* **CGQ improves value estimation over single-step and action-chunked TD learning.** We compare single-step TD, action-chunked TD, and a simple convex combination of the two (equation 19) in a simple gridworld using a fixed offline dataset. To simulate function approximation error, we add noise to TD targets. The upper-right panel shows the optimal value function. The left panels visualize the value prediction error after $k \in \{1, 3, 10, 100\}$ TD updates and the middle panels show the learned value functions. Single-step TD exhibits large errors in states far from the goal due to bootstrapping error accumulation over long backup chains, while action-chunked TD propagates values quickly but can converge to a suboptimal critic due to open-loop chunks, leaving persistent errors at intermediate states. In contrast, a convex combination of the two, a conceptual proxy for CGQ, achieves lower error (see the lower-right plot) by combining rapid chunked value propagation with fine-grained single-step value backups.

**FQL with action chunks (QC-FQL).** FQL naturally extends to action-chunked Q-learning by operating on action sequences instead of single actions. QC-FQL (Li et al., 2025b) learns an action-chunked policy and critic with chunk length $h$. Analogous to FQL, QC-FQL first fits a flow-based behavior policy $\pi_{\theta_c}(s, \mathbf{z}) : \mathcal{S} \times \mathcal{A}^h \to \mathcal{A}^h$ by learning a velocity field $v_{\theta_c}$:

$$\mathcal{L}_c^{\text{flow}}(\theta_c) = \mathbb{E}_{\substack{(s_t, \mathbf{a}_t) \sim \mathcal{D}, \\ \mathbf{z} \sim \mathcal{N}(0, I_{\mathcal{A}^h}), \\ u \sim \mathrm{U}([0,1])}} \left[ \|v_{\theta_c}(u, s_t, \mathbf{a}_z) - (\mathbf{a}_t - z)\|_2^2 \right],$$

$$(5)$$

where $\mathbf{a}_z = (1 - u)z + u\mathbf{a}_t$. QC-FQL then learns a constrained action-chunked policy $\pi_{\omega_c}(s, \mathbf{z})$ by optimizing

$$\mathcal{L}_c^{\text{FQL}}(\omega_c) = \mathbb{E}_{s \sim \mathcal{D}, \mathbf{z} \sim \mathcal{N}(0, I_{\mathcal{A}})} \big[ -Q_{\phi_c}(s, \pi_{\omega_c}(s, \mathbf{z})) + \alpha \|\pi_{\theta_c}(s, \mathbf{z}) - \pi_{\omega_c}(s, \mathbf{z})\|_2^2 \big].$$

$$(6)$$

In this paper, we use QC-FQL for action-chunked policy extraction.

# 3. Motivation: Trade-offs in Action-Chunked TD Learning

Before introducing our method, we examine the complementary strengths and weaknesses of single-step and action-chunked TD learning, which our approach is designed to balance. Figure 2 illustrates this trade-off in a simple gridworld. Single-step TD learning eventually converges to the correct value function, but requires many updates to propagate reward signal, making it vulnerable to bootstrapping error accumulation in long-horizon tasks. Action-chunked TD learning propagates values rapidly in early iterations, but its open-loop backups can fail to stitch across action

chunks, leading to a suboptimal value function even after convergence.

## 3.1. Action-Chunked TD Learning Reduces Bootstrapping Error Accumulation

Single-step TD learning is vulnerable to bootstrapping error accumulation since the TD target, $r_t + \gamma Q(s_{t+1}, a_{t+1})$, depends on the critic's own predictions. Errors in value estimates at future states can therefore propagate backward through successive Bellman updates. This issue is exacerbated as the horizon increases and reward information must be propagated through many consecutive bootstrapping steps.

Action-chunked TD learning partially mitigates this issue by reducing the effective horizon length (backup depth). For chunk size $h$, letting $\mathbf{a}_t = (a_t, a_{t+1}, \cdots, a_{t+h-1})$, the chunked TD target takes the form:

$$\sum_{k=0}^{h-1} \gamma^k r_{t+k} + \gamma^h Q(s_{t+h}, \mathbf{a}_{t+h}), \qquad (7)$$

which requires fewer bootstrapping steps to propagate reward information across long horizons. Although the target still depends on the critic, the error accumulation term scales with $\frac{1}{1-\gamma^h}$ rather than $\frac{1}{1-\gamma}$, which can yield up to an $h$-fold improvement when $\gamma \approx 1$ (see Theorem 4.1). This reduction helps explain the fast initial value propagation observed for chunk-based TD learning in Figure 2.

## 3.2. Action-Chunked Critics Can Be Suboptimal

Despite its improved stability, action chunking can be structurally suboptimal relative to single-step TD learning be-

cause it restricts the policy class to open-loop action sequences. Let $Q^*(s_t, a_t)$ denote the optimal single-step Q-function and let $Q_c^*(s_t, \mathbf{a}_t)$ as the optimal Q-function for an action-chunked policy. Since action chunking restricts the policy class to open-loop action sequences, in general

$$Q_c^*(s_t, \mathbf{a}_t) \leq Q^*(s_t, a_t). \qquad (8)$$

In stochastic environments where optimal behavior is reactive, this restriction can prevent precise action stitching across states, leading to persistent value inaccuracies at intermediate states. This effect is visible in Figure 2: despite reduced early propagation, chunk-based backups can converge to a value function that remains suboptimal in parts of the state space.

**Implication.** Taken together, these observations suggest a fundamental trade-off. Action-chunked critics provide more stable long-horizon value propagation by reducing backup depth, but can sacrifice optimality due to open-loop execution. Single-step TD learning supports fine-grained value updates and trajectory stitching, but is more susceptible to bootstrapping error accumulation over long horizons. This complementary structure motivates combining the two: *leverage chunked learning signals for stability while retaining single-step structure for compositionality.*

## 4. Chunk-Guided Q-Learning

We introduce **Chunk-Guided Q-Learning (CGQ)**, a simple offline RL method that combines the long-horizon stability of action-chunked TD learning with the fine-grained optimality of single-step TD learning. CGQ regularizes the single-step critic toward a chunked critic trained with temporally extended backups, reducing compounding error while preserving step-wise updates.

We emphasize that *our goal is to train a single-step policy and critic*; the action-chunked policy and critic are trained only as auxiliary components to guide learning of the single-step critic.

Throughout this paper, we use $\phi$ to denote critic parameters and $\theta, \omega$ to denote policy parameters. We use the subscript $(\cdot)_c$ to indicate quantities associated with action-chunked critics, policies, or losses.

### 4.1. Practical Implementation

**Action-chunked TD learning.** We define an action-chunked critic $Q_{\phi_c}(s_t, \mathbf{a}_t)$ that evaluates an $h$-step action sequence $\mathbf{a}_t = (a_t, \ldots, a_{t+h-1})$. Given an action-chunked policy $\pi_{\omega_c}(\mathbf{a}_t \mid s_t)$, we train the action-chunked critic $Q_{\phi_c}$

with the action-chunked TD loss:

$$\mathcal{L}_c^{\text{TD}}(\phi_c) = \mathbb{E}_{(s_t, \mathbf{a}_t) \sim \mathcal{D}} \Big[ \big( Q_{\phi_c}(s_t, \mathbf{a}_t) - \mathbf{r}_t \\ - \gamma^h Q_{\bar{\phi}_c}(s_{t+h}, \mathbf{a}_{t+h}) \big)^2 \Big], \qquad (9)$$

where $\mathbf{r}_t = \sum_{k=0}^{h-1} \gamma^k r_{t+k}$ and $\mathbf{a}_{t+h} \sim \pi_{\omega_c}(\cdot \mid s_{t+h})$.

While the action-chunked critic can be suboptimal, it typically reduces bootstrapping error accumulation. We optimize $\pi_{\omega_c}$ using QC-FQL (Li et al., 2025b) for its simplicity and robustness across tasks.

**Single-step TD learning.** We train a single-step critic $Q_\phi(s, a)$ with the standard TD objective:

$$\mathcal{L}^{\text{TD}}(\phi) = \mathbb{E}_{\substack{(s,a,r',s') \sim \mathcal{D}, \\ a' \sim \pi_\omega(\cdot|s')}} \Big[ \big( Q_\phi(s, a) - r - \gamma Q_{\bar{\phi}}(s', a') \big)^2 \Big], \qquad (10)$$

where $\bar{\phi}$ is a target network updated as an exponential moving average of $\phi$ (Mnih et al., 2013). Single-step TD enables fine-grained value propagation across states, but suffers from bootstrapping error accumulation over long horizons.

**Critic regularization via action-chunked critic.** To combine the complementary strengths of both critics, CGQ regularizes the single-step critic $Q_\phi(s_t, a_t)$ toward the action-chunked critic $Q_{\phi_c}(s_t, \mathbf{a}_t)$ using an *upper-expectile* distillation loss:

$$\mathcal{L}^{\text{reg}}(\phi) = \mathbb{E}_{(s_t, \mathbf{a}_t) \sim \mathcal{D}} \Big[ \ell_\tau \big( Q_{\phi_c}(s_t, \mathbf{a}_t) - Q_\phi(s_t, a_t) \big) \Big]. \qquad (11)$$

where $\ell_\tau(u) = |\tau - \mathbb{I}(u < 0)| u^2$, with $0.5 \leq \tau < 1$ (Kostrikov et al., 2022). This asymmetric alignment biases the single-step critic toward higher-value estimates from the chunked critic, allowing the single-step critic to be guided by the action-chunked critic, while being less affected from the suboptimality of the action-chunked critic.

**Overall objective.** We train the single-step critic $Q_\phi$ by minimizing

$$\mathcal{L}^{\text{CGQ}}(\phi) = \mathcal{L}^{\text{TD}}(\phi) + \beta \mathcal{L}^{\text{reg}}(\phi), \qquad (12)$$

where $\beta$ controls the strength of guidance from the action-chunked critic. We then extract the single-step policy $\pi_\omega$ from $Q_\phi$ using the FQL policy extraction (Park et al., 2025d). The overall algorithm is described in Algorithm 1.

### 4.2. Theoretical Analysis

We now provide a theoretical analysis of CGQ to formalize the trade-off introduced by chunk guidance. In particular, we study how regularizing single-step TD learning toward an action-chunked critic affects the optimality and error propagation of the learned value function.

---

**Algorithm 1** Chunk-Guided Q-Learning (CGQ)

---

**Require:** Offline dataset $\mathcal{D}$, action chunk size $h$
 1: Initialize single-step critic $Q_\phi$, behavioral flow policy $\pi_\theta$, policy $\pi_\omega$
 2: Initialize action-chunked critic $Q_{\phi_c}$, behavioral flow policy $\pi_{\theta_c}$, policy $\pi_{\omega_c}$
 3: **while** not converged **do**
 4:     Sample $\{(s_t, \mathbf{a}_t, r_{t:t+h}, s_{t+1}, s_{t+h})\} \sim \mathcal{D}$
 5:     ▷ Train action-chunked policy $\pi_{\omega_c}$ and critic $Q_{\phi_c}$
 6:     Update chunked critic $Q_{\phi_c}$ to minimize $\mathcal{L}_c^{\text{TD}}(\phi_c)$
 7:     Update chunked policies $\pi_{\theta_c}, \pi_{\omega_c}$ with Eqs. (5) and (6)
 8:     ▷ Train single-step policy $\pi_\omega$ and critic $Q_\phi$
 9:     Update critic $Q_\phi$ to minimize $\mathcal{L}^{\text{TD}}(\phi) + \beta \mathcal{L}^{\text{reg}}(\phi)$
10:     Update policies $\pi_\theta, \pi_\omega$ with Eqs. (3) and (4)
11: **end while**
12: **return** single-step policy $\pi_\omega$

---

Minimizing the one-step Bellman residual $\|\mathcal{T}Q - Q\|$ should recover the optimal action-value function $Q^*$, since $Q^*$ is the unique fixed point of the Bellman operator $\mathcal{T}Q(s,a) = \mathbb{E}[r(s,a) + \gamma \max_{a'} Q(s',a')]$. In practice, however, the Bellman evaluation is noisy due to function approximation, finite data coverage, and other sources of error. To model this behavior, we consider a stochastic iterative process defined as

$$\widehat{\mathcal{T}}Q_k = \mathcal{T}Q_k + \varepsilon_k, \tag{13}$$

where $\varepsilon_k$ is a zero-mean random perturbation satisfying $\mathbb{E}[\|\varepsilon_k\|^2] \leq \sigma^2$ for some $\sigma > 0$.

Let $Q^*$ denote the fixed point of $\mathcal{T}$. Under a contraction assumption of operator $\mathcal{T}$, we obtain the following asymptotic error bound.

**Theorem 4.1** (**Error accumulation of one-step TD learning**). *Let $\mathcal{T}$ be a $\gamma$-Lipschitz linear operator in $L_2$ norm, and $\widehat{\mathcal{T}}$ be a stochastic iterative process defined as above. Then, the asymptotic expected squared error satisfies:*

$$\limsup_{k \to \infty} \mathbb{E}[\|Q_k - Q^*\|^2] \leq \frac{\sigma^2}{1 - \gamma^2} \tag{14}$$

Theorem 4.1 indicates that the error in each Bellman update accumulates geometrically, and therefore the overall error scales with $\mathcal{O}(\sqrt{1/(1 - \gamma^2)})$. In long-horizon tasks, where $\gamma$ is close to 1, this error accumulation exacerbates.

Here, we prove how CGQ can improve this bound by adding a regularization towards the suboptimal but stable action-chunked critic. We note that for the following derivations, the regularization target does not need to be an action-chunked critic in general; however, we keep the notation of regularization target as $Q_c$, as we focus on the action-

chunked critic in our method. To control interactions between stochastic noise and regularization bias, we additionally assume that the curvature of $\mathcal{T}$ is bounded.

**Theorem 4.2** (**Improved bound via CGQ regularization, informal**). *Let $\widehat{\mathcal{T}}_\beta$ be the regularized stochastic iterative process: $\widehat{\mathcal{T}}_\beta Q_k = \frac{1}{1+\beta} \widehat{\mathcal{T}} Q_k + \frac{\beta}{1+\beta} Q_c$. Then, the asymptotic expected squared error satisfies:*

$$\limsup_{k \to \infty} \mathbb{E}[\|Q_k - Q^*\|^2]$$
$$\leq \underbrace{\frac{\sigma^2}{(1+\beta)^2 - \gamma^2}}_{\text{Error accumulation}} + \underbrace{\frac{\beta^2 \|Q^* - Q_c\|^2}{(1+\beta-\gamma)^2}}_{\text{Suboptimality}} \tag{15}$$
$$+ \underbrace{\frac{L(1+\beta)\beta\|Q^* - Q_c\|\sigma^2}{((1+\beta)^2 - \gamma^2)(1+\beta-\gamma)^2}}_{\text{Cross-term}}$$

*where $L$ denotes the maximum curvature of $\mathcal{T}$.*

*This bound is minimized at some $\beta^* < \infty$, being strictly smaller than $\|Q^* - Q_c\|^2$. Moreover, if (but not necessarily only if) $L\|Q_c - Q^*\| < \frac{2(1-\gamma)}{1+\gamma}$, this bound is minimized at some $0 < \beta^* < \infty$, and is strictly smaller than both $\frac{\sigma^2}{1-\gamma^2}$ and $\|Q^* - Q_c\|^2$.*

Theorem 4.2 makes explicit the bias–variance trade-off induced by chunk regularization. The first term captures error accumulation from noisy single-step TD updates, the second reflects bias from regularization toward a suboptimal critic, and the third term quantifies their interaction. The theorem shows that CGQ always gives tighter bound than action-chunked TD learning.

Importantly, whether an interior optimum exists (i.e., whether CGQ outperforms pure single-step TD learning) depends on the magnitude of the cross-term, which is determined by the curvature $L$ of the Bellman operator. We emphasize that this curvature condition is sufficient but not necessary; it can still have a nonzero finite minimizer $\beta^*$ with strictly smaller bound, even if the condition is not satisfied. Please see Section E for the formal statement and the proof of Theorem 4.2.

## 5. Experiments

In this section, we evaluate CGQ on challenging long-horizon tasks and compare against strong single-step and action-chunked offline RL methods. We further analyze when chunk guidance helps via ablations on the regularization strength and chunk length.

### 5.1. Experimental Setup

We evaluate CGQ on the OGBench benchmark (Park et al., 2025b), which contains challenging long-horizon

*Table 1.* **Results in OGBench manipulation and navigation benchmarks.** Entries report average success rate over 150 evaluation episodes, aggregated across 4 seeds. We report mean and standard deviation within each task category when available; entries without "±" are copied from prior work (Park et al., 2025d; Li et al., 2025b; Kim et al., 2025). We highlight results within 95% of the best performance. We note that DQC is evaluated under smaller, reward-based datasets in our setting (rather than the large goal-conditioned datasets used in the original paper), which may lead to lower performance than originally reported.

| Task Category | FQL | NFQL | QC-FQL | DEAS | DQC | CGQ |
|---|---|---|---|---|---|---|
| scene-sparse (5 tasks) | 57 | 18 | 84 | $93_{\pm2}$ | $81_{\pm2}$ | $86_{\pm3}$ |
| cube-double (5 tasks) | 29 | 11 | 39 | 48 | $31_{\pm2}$ | $69_{\pm3}$ |
| puzzle-3x3-sparse (5 tasks) | 100 | 98 | 63 | $99_{\pm1}$ | $100_{\pm0}$ | $99_{\pm3}$ |
| puzzle-4x4 (5 tasks) | 17 | $23_{\pm5}$ | $26_{\pm2}$ | $39_{\pm3}$ | $8_{\pm2}$ | $28_{\pm2}$ |
| cube-triple (5 tasks) | 10 | 23 | 83 | 82 | $2_{\pm1}$ | $94_{\pm1}$ |
| cube-quadruple (5 tasks) | 17 | 36 | 45 | 64 | $56_{\pm22}$ | $82_{\pm2}$ |
| Average (Manipulation) | 38 | 35 | 57 | 71 | $46_{\pm3}$ | $76_{\pm2}$ |
| antmaze-large (5 tasks) | 79 | $47_{\pm5}$ | $20_{\pm3}$ | $67_{\pm3}$ | $71_{\pm6}$ | $67_{\pm3}$ |
| antmaze-giant (5 tasks) | 9 | $2_{\pm1}$ | $0_{\pm0}$ | $8_{\pm1}$ | $10_{\pm1}$ | $4_{\pm2}$ |
| humanoidmaze-medium (5 tasks) | 58 | $23_{\pm3}$ | $4_{\pm1}$ | $37_{\pm3}$ | $93_{\pm2}$ | $34_{\pm5}$ |
| Average (Navigation) | 49 | $24_{\pm3}$ | $8_{\pm2}$ | $37_{\pm1}$ | $58_{\pm3}$ | $35_{\pm4}$ |

tasks. For robotic manipulation, we consider six tasks: `cube-double`, `cube-triple`, `cube-quadruple`, `puzzle-3x3`, `puzzle-4x4`, and `scene`, and train on the `play` dataset. For navigation, we evaluate three tasks: `antmaze-large`, `antmaze-giant`, and `humanoidmaze-medium`, and train on the `navigate` dataset. Additional training details are provided in Appendix, Section A.

**Methods.** We compare against both single-step and action-chunked offline RL methods. For single-step methods, we consider FQL (Park et al., 2025d), which learns a behavior flow policy and regularizes the one-step policy with the behavior policy, and its variant with $n$-step return, denoted as NFQL. For action-chunked methods, we consider QC (Li et al., 2025b), DEAS (Kim et al., 2025), and DQC (Li et al., 2025a). QC adapts action-chunked critic and policy to existing single-step frameworks (e.g., FQL). DEAS decouples policy and value learning using expectile regression, following IQL (Kostrikov et al., 2022). DQC decouples chunk lengths for the critic and policy by distilling a large-chunk critic into a smaller-chunk critic for the policy.

### 5.2. Results

We report the performance of CGQ and baselines in Table 1 on OGBench manipulation and navigation tasks.

**Manipulation.** On manipulation tasks, CGQ achieves the best performance across all task categories, outperforming single-step, $n$-step, and action-chunked offline RL methods. Notably, FQL performs well on relatively shorter-horizon tasks such as `puzzle-3x3-sparse` and `scene-sparse`,

but degrades significantly on longer-horizon tasks including `puzzle-4x4`, `cube-triple`, and `cube-quadruple`. In contrast, action-chunked methods struggle on shorter-horizon tasks (e.g., `puzzle-3x3-sparse`) due to the restricted open-loop policy class. CGQ substantially outperforms in both regimes while using a single-step policy, indicating that chunking guidance successfully balances accurate step-wise value propagation with reduced bootstrapping error.

**Navigation (negative results).** On navigation tasks, CGQ outperforms or matches methods that use action-chunked policies, but underperforms FQL, which does not rely on an action-chunked critic. This is consistent with prior observations that action-chunked methods can be less effective in locomotion domains (Park et al., 2025a; Li et al., 2025a), where highly reactive control and fine-grained value estimation are crucial. We believe this is primarily due to the difficulty of action-chunked policy learning in locomotion, which in turn yields an unreliable chunked critic that corrupts the single-step critic via regularization. While DEAS employs IQL-style critic learning, it also suffers from this issue. DQC partially avoids this by decoupling the chunk sizes for the critic and the policy; notably, our empirical tuning selects a policy chunk size of 1 for all navigation tasks, thereby reducing the burden of chunked policy learning. Improving the chunked critic and developing adaptive weighting schemes for the regularizer are promising directions for future work.

### 5.3. Q&As

**Q: How is CGQ different from n-step methods, which also use single-step policy and critic?**

While CGQ and $n$-step methods both learn a single-step policy and critic, they differ fundamentally in how they incorporate horizon reduction. In $n$-step methods, horizon reduction is achieved by modifying the one-step TD target with multi-step returns. However, $n$-step targets still bootstrap from the same single-step critic at the tail, so biased critic estimates can feed back into the target and perpetuate the very error accumulation that $n$-step TD was meant to address.

In contrast, CGQ introduces horizon reduction via an auxiliary action-chunked critic that is trained independently from the single-step critic and never directly enters the TD target. Rather than modifying the bootstrapping chain, the action-chunked critic serves as an external regularizer that guides the single-step critic, avoiding additional error propagation. Consistent with this distinction, NFQL, an $n$-step variant of FQL, performs substantially worse than CGQ (Table 1).

**Q: Is CGQ the best way to blend action-chunked RL with single-step RL?**

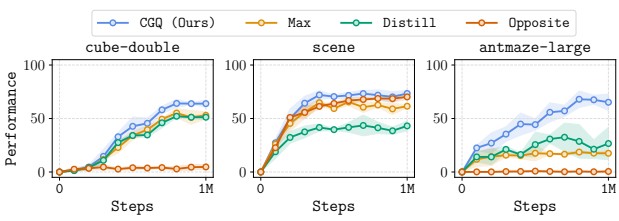

*Figure 3.* **Performance of various TD update designs for blending single-step and multi-step TD learning.** CGQ's regularization yields the most effective critic.

To investigate this question, we evaluate several alternative designs for combining action-chunked and single-step value learning, and compare them against CGQ.

**CGQ-DISTILL.** This variant removes single-step TD learning and trains the critic using only the distillation loss, i.e., $\mathcal{L}(\phi) = \mathcal{L}^{\text{reg}}$, relying entirely on chunk-based value backups. It is analogous to DQC (Li et al., 2025a) with $h_a = 1$, and tests whether chunk guidance alone is sufficient for accurate value estimation.

**CGQ-MAX.** This variant constructs an optimistic target by taking the maximum of the single-step and chunk-based value estimates:

$$\mathcal{L}^{\max}(\phi) = \mathbb{E}_{s,a,r,s',\mathbf{a} \sim \mathcal{D}, a' \sim \pi_\omega} \Big[ \Big( Q_\phi(s,a) - \max \big( r + \gamma \bar{Q}_\phi(s',a'), Q_{\phi_c}(s,\mathbf{a}) \big) \Big)^2 \Big]. \quad (16)$$

This tests whether aggressively following the larger of the single-step and chunk-based estimates improves performance.

**CGQ-OPPOSITE.** This variant regularizes the action-chunked critic toward a single-step critic. While superficially symmetric, it takes the worst side of both; the chunked critic remains constrained by open-loop execution, and guidance from the single-step critic reintroduces sensitivity to bootstrapping errors.

As shown in Figure 3, none of these alternatives matches CGQ. While CGQ is not necessarily the optimal or unique way to combine action-chunked and single-step RL, these results suggest that CGQ's simple regularization scheme provides an effective balance between reduced error accumulation and fine-grained single-step value propagation.

**Q: How does action-chunk size affect CGQ?**

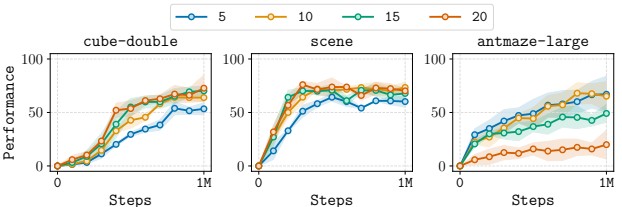

*Figure 4.* **Performance of CGQ across different action-chunk sizes $h$.** In general, larger chunks improve performance.

We analyze the effect of the chunk size $h$ for the action-chunked TD learning by evaluating CGQ with $h \in \{5, 10, 15, 20\}$ on cube-double, scene, and antmaze-large. For the manipulation tasks, cube-double and scene, larger chunks generally improve performance, suggesting that stronger horizon reduction yields better regularization and improved performance where chunked TD learning is reliable. In contrast, on the navigation task antmaze-large, performance degrades as the chunk size increases. This trend aligns with the results in Table 1: learning action-chunked policies is substantially more difficult in locomotion domains, making the chunk-based critic less reliable for regularization.

**Q: How important is the distillation coefficient $\beta$?**

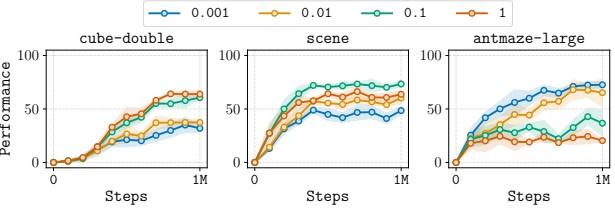

*Figure 5.* **Performance of CGQ under varying $\beta$.** Choice of $\beta$ is important for the performance.

We investigate the importance of the distillation coefficient $\beta$, which controls the strength of regularization towards the action-chunked critic. Specifically, we evaluate CGQ

with $\beta \in \{0.001, 0.01, 0.1, 1\}$ and report the result in Figure 5. We find that the preferred value of $\beta$ depends on how reliable the chunk-based critic is. In manipulation tasks, stronger regularization is beneficial. This indicates that the chunked critic provides useful guidance for value learning. In contrast, `antmaze-large` favors smaller $\beta$, suggesting that the chunked critic is less reliable in locomotion domains and should therefore be used more conservatively.

**Q: Is expectile coefficient $\tau$ important?**

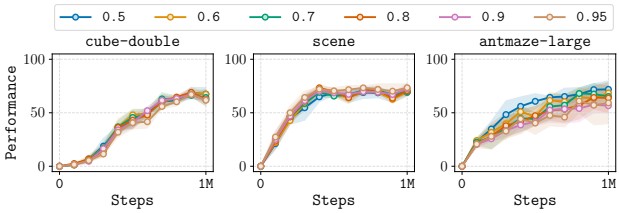

*Figure 6.* **Performance of CGQ under varying expectile coefficient $\tau$.** CGQ is robust to choice of $\tau$.

We study the effect of the expectile parameter $\tau$ used in the distillation loss across three tasks, using $\tau \in \{0.5, 0.6, 0.7, 0.8, 0.9, 0.95\}$. As shown in Figure 6, CGQ is robust to the choice of $\tau$: the performance is similar even when using $\tau = 0.5$ (i.e., without optimistic weighting), in contrast to prior observations in DQC (Li et al., 2025a). We attribute this robustness CGQ using the action-chunked critic as a stabilizing regularizer, rather than relying on pure distillation, as in DQC.

## 6. Related Work

**Offline RL.** Offline RL aims to learn a policy from a fixed dataset without further environment interaction. The main challenge is distributional shift (Levine et al., 2020), maximizing return often pushes the learned policy away from the data distribution, making value estimation unreliable. Prior work has explored several strategies have been explored to mitigate this issue, including behavior regularization (Wu et al., 2019; Peng et al., 2019; Fujimoto & Gu, 2021; Tarasov et al., 2023a; Park et al., 2025d), conservative value estimation (Kumar et al., 2020), in-sample maximization (Kostrikov et al., 2022; Xiao et al., 2023), rejection sampling (Chen et al., 2023; Hansen-Estruch et al., 2023), and other approaches such as sequence-modeling (Chen et al., 2021; Janner et al., 2021; Jiang et al., 2023) or model-based methods (Kidambi et al., 2020; Yu et al., 2020; 2021; Liu et al., 2023b), and more (Eysenbach et al., 2022; Wang et al., 2023; An et al., 2021; Brandfonbrener et al., 2021; Myers et al., 2025; Park et al., 2025e).

Despite these advances, offline RL remains challenging in long-horizon tasks due to error accumulation: small value estimation errors can compound across successive Bellman backups when regression targets depend on the model's own predictions (Park et al., 2025c; Sutton et al., 1998). Recent work has shown that *horizon-reduction* techniques can effectively mitigate bootstrapping error accumulation by shortening the effective backup depth, including action chunking (Li et al., 2025b;a), $n$-step returns (Park et al., 2025c), and hierarchical RL (Park et al., 2024; 2025c). Our method follows this line of work but introduces a novel approach that balances single-step TD learning with horizon-reduced TD learning, combining their complementary strengths.

**Action chunking for offline RL.** Originally popularized in imitation learning (Zhao et al., 2023; Chi et al., 2023; Kim et al., 2024), action chunking has recently emerged as a practical horizon-reduction strategy for offline RL (Seo & Abbeel, 2025; Li et al., 2025b; Kim et al., 2025; Li et al., 2025a). In this framework, the policy predicts a sequence of actions $\mathbf{a}_t = (a_t, \ldots, a_{t+h})$, and the critic evaluates the value of an entire action chunk, $Q(s_t, \mathbf{a}_t)$, rather than a single action. By reducing the number of TD backups (by a factor of $h$), these methods improve robustness to the bias introduced by self-bootstrapping during value propagation.

However, action chunking imposes several challenges in offline RL. For example, by enlarging the action space, action chunking complicates policy optimization and exacerbates value overestimation. Prior work mitigates these issues through different mechanisms: QC (Li et al., 2025b) constrains policies to limit distributional shift via FQL-based extraction (Park et al., 2025d) or rejection sampling (Chen et al., 2023); DEAS (Kim et al., 2025) decouples policy and value learning using expectile regression, following IQL (Kostrikov et al., 2022); and DQC (Li et al., 2025a) highlights the difficulty of predicting long action sequences and instead performs implicit value learning over large chunks, and distill into a critic with a smaller chunk. However, action-chunked critics have a fundamental limitation: they fail to account for reactive policies, leading to suboptimal value estimation even if perfectly optimized. Addressing this limitation is the focus of this paper.

While both DQC and CGQ leverage a chunk-based critic alongside a finer-grained critic, the two methods differ fundamentally in motivation and mechanism. DQC focuses on policy extraction: the smaller-chunk critic is trained solely via distillation from the chunk-based critic, without any finer-grained TD updates, and thus inherits the suboptimality of open-loop chunked value learning. CGQ instead targets value learning itself, training the single-step critic with standard single-step TD learning and using the chunk critic only as a regularization signal, achieving both the stability of chunked backups and the optimality of single-step value learning.

## 7. Closing Remarks

We presented Chunk-Guided Q-Learning (CGQ), a single-step offline RL method that stabilizes value estimation by regularizing TD learning toward an auxiliary chunk-based critic. CGQ is designed to combine the complementary strengths of single-step and chunk-based TD learning: chunk-based value backups provide stable long-horizon learning signals with reduced compounding error, while single-step TD learning preserves critic optimality and supports reactive policy learning. Theoretically, we formalize the trade-off between the error accumulation of single-step TD learning and the potential suboptimality of chunk-based TD learning, and show that CGQ can find a sweet spot that can improve over either extreme. Empirically, CGQ achieves strong performance on challenging long-horizon OGBench tasks, often outperforming both single-step and action-chunked methods.

CGQ also has limitations. When the action-chunked critic is inaccurate, the guidance may become not reliable and purely single-step methods may perform better. We provide an initial discussion of this reliability issue, including simple Q-value and Q-loss filtering variants, in Section D. Moreover, CGQ depends on the regularization weight $\beta$ and selecting $\beta$ can be challenging when transferring to new tasks or multi-task settings. Promising directions for future work include improving the chunk-based guidance signal and developing adaptive or principled strategies for tuning $\beta$.

## Acknowledgements

This research was supported by the National Research Foundation of Korea (NRF) grant (RS-2024-00333634), the Institute of Information & Communications Technology Planning & Evaluation (IITP) grants (RS-2020-II201361, Artificial Intelligence Graduate School Program (Yonsei University); and RS-2024-00509279, Global AI Frontier Lab) funded by the Korean government (MSIT).

## Impact Statement

This work aims to improve long-horizon offline learning for autonomous agents, e.g., robots and self-driving cars. While the methods could be used in safety-critical settings, we do not anticipate immediate negative societal impacts from this paper.

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

# A. Training Details

We implement CGQ on top of the codebase of (Li et al., 2025b). Each experiment tasks around 3 hours and approximately 18 hours for `cube-triple`, `cube-quadruple` tasks on a single RTX3090 GPU.

## A.1. Benchmark and Datasets

In this work, we consider 3 navigation tasksets and 6 manipulation tasksets from OGBench (Park et al., 2025b) and use the `singletask` versions in our experiments (45 tasks in total). Reward setup in the navigation domain is sparse: a reward of -1 is given unless the task is complete, otherwise 0. For `scene` and `puzzle-3x3` domain in manipulation, we sparsify the rewards following(Li et al., 2025b). For `cube-double`, `cube-triple`, `cube-quadruple`, `puzzle-4x4`, reward is calculated according to the number of completed subtasks (e.g., the number of correctly placed cubes or the number of correctly matched buttons) (Park et al., 2025b).

*Table 2.* **Metadata of each task categories used in OGBench experiments.** We report the dataset size, episode length, and action dimensionality for all 9 task sets used in our experiments. For the harder tasks, such as `cube-triple` and `cube-quadruple`, we use larger 10M/100M-sized dataset.

| Task Category | Dataset Size | Episode Length | Action Dimension) |
|---|---|---|---|
| antmaze-large | 1M | 1000 | 8 |
| antmaze-giant | 1M | 2000 | 8 |
| humanoidmaze-medium | 2M (default) | 2000 | 21 |
| scene-sparse | 1M | 750 | 5 |
| puzzle-3x3-sparse | 1M | 500 | 5 |
| cube-double | 1M | 500 | 5 |
| puzzle-4x4 | 1M | 500 | 5 |
| cube-triple | 10M | 1000 | 5 |
| cube-quadruple | 100M | 1000 | 5 |

## A.2. Training and Evaluation Protocol

We train CGQ for 1M gradient steps on the 1M and 2M datasets, and for 2.5M gradient steps on the 10M and 100M datasets for state-based OGBench tasks. For D4RL and pixel-based OGBench tasks, we train with 500K gradient steps. All agents are evaluated every 100K gradient steps using 50 evaluation episodes.

Following Kim et al. (2025), the 10M dataset is constructed from the first 10 files of the 100M dataset. For OGBench tasks, we follow Park et al. (2025c) and report the average success rate over the last three evaluation epochs: 800K/900K/1M for 1M-step state-based training, 2.3M/2.4M/2.5M for 2.5M-step state-based training, and 300K/400K/500K for pixel-based training. For D4RL, following Tarasov et al. (2023b), we report the performance at the last epoch. For offline-to-online RL experiments, we report performance after 1M and 2M steps.

**Results from prior works.** For tasks that are already evaluated in prior work, we use the reported results (i.e., numbers without ± in Table 1). Specifically, for manipulation tasks, results of `cube-double`, `scene-sparse`, and `puzzle-3x3-sparse` are taken from (Li et al., 2025b), except for DEAS on `cube-double`, where results are from (Kim et al., 2025). Results for `cube-triple` and `cube-quadruple` are also obtained from (Kim et al., 2025). For navigation tasks, FQL results are taken from (Park et al., 2025d). Otherwise, we implement the baselines in our codebase and evaluated according to our evaluation protocol.

## A.3. Training Time

We report per-step training and inference time of all methods on a single L40S GPU in Table 3. CGQ's training cost is approximately $2.3\times$ that of FQL, as CGQ jointly trains both a single-step and an action-chunked critic. Inference cost remains comparable to single-step baselines, as only the single-step policy is used at test time.

*Table 3.* Per-step run time comparison on a single L40S GPU.

|  | FQL | NFQL | QC-FQL | DEAS | DQC | CGQ |
|---|---|---|---|---|---|---|
| Train (ms) | 2.00 | 2.14 | 2.11 | 2.44 | 2.97 | 4.66 |
| Inference (ms) | 0.26 | 0.27 | 0.26 | 0.29 | 0.66 | 0.36 |

## A.4. Implementation Details

**Network architecture.** Following FQL (Park et al., 2025d), we use [512, 512, 512, 512] 4-layer MLPs with layer normalization (Ba et al., 2016) for critic and policy networks. For `cube-triple` and `cube-quadruple` tasks, which we use larger datasets, we increased the model size to [1024, 1024, 1024, 1024] for all neural networks.

**Value learning.** For both 1-step critic and chunked critic, we train two Q functions and take the mean of the two Q values for the Q value in the actor objective, calculating the target for the TD learning and the regularization. to improve stability.

## A.5. Baselines.

Here, we briefly introduce the baselines used in this paper.

**FQL** (Park et al., 2025d) is a behavior regularization-based offline RL method that learns a constrained policy from an expressive flow-based behavior policy. Despite its simplicity, FQL serves as a strong baseline for various offline RL benchmarks (e.g., OGBench). In this paper, we also use FQL as a policy extraction method for the single-step policy.

**QC-FQL** (Li et al., 2025b) extends FQL to action-chunked Q-learning by operating on action sequences instead of single actions. QC-FQL improves FQL's performance in manipulation tasks, especially in challenging tasks such as `cube-triple` or `cube-quadruple`. In this paper, we also use QC-FQL as a policy extraction method for the action-chunked policy.

**DEAS** (Kim et al., 2025) extends action-chunked offline RL by incorporating detached value learning (Kostrikov et al., 2022) which decouples critic training from the policy to prevent value overestimation in the expanded action space. DEAS further combines distributional RL objectives and dual discount factors to mitigate the high variance introduced by the $n$-step returns.

**N-FQL** is an $n$-step return variant of FQL that incorporates horizon reduction via multi-step returns, while retaining a single-step critic and policy.

**DQC.** decouples the chunk sizes used by the critic and the policy by distilling an action-chunked critic into a critic with a smaller chunk size used by the policy. While DQC employs a binary cross-entropy loss for value learning under a goal-conditioned setting, we utilize mean-squared error loss instead, as rewards generally lie in the range $[-N, 0]$ in our experimental setup. We use the official implementation only with this necessary change.

## A.6. Gridworld Experimental Details

We provide the details for the gridworld experiment used in Figure 2.

**Environment and dataset generation.** The environment consists of an $18 \times 18$ discrete grid. The agent's objective is to reach a fixed goal state located at coordinates $(6, 8)$.

- **Dynamics:** The agent can move in four cardinal directions (up, down, left, right). Transitions that would lead out of the grid boundaries makes the agent stay in its current state.

- **Reward Function:** A sparse reward of 1 is given only when the agent enters the goal state. For all other transitions, the reward is 0. The episode is terminated when the agent reaches the goal. We use a discount factor of $\gamma = 0.9$.

- **Offline Dataset:** We collect a dataset $\mathcal{D}$ consisting of 60 trajectories, each with a maximum length of 15 steps. To simulate a suboptimal, exploratory data distribution, trajectories are generated using an $\epsilon$-greedy sampling ($\epsilon = 0.9$) from the optimal policy.

**Value Backup Definitions** We compare three distinct backup operators to analyze the efficiency of value propagation over iteration $k$.

**1. Single-step Value Backup** The standard 1-step backup updates the value function using individual transitions $(s, s')$ available in the dataset:

$$V_{k+1}(s) \leftarrow \max_{s' \in \mathcal{D}(s)} \left[ R(s, s') + \gamma V_k(s') \right] \tag{17}$$

**2. Chunk-based Value Backup ($h$-step)** The chunk-based operator utilizes a sequence of states $(s_t, s_{t+1}, \ldots, s_{t+h})$ of length $h$. The backup is defined as:

$$V_{k+1}(s_t) \leftarrow \max_{\text{chunk} \in \mathcal{D}} \left[ \mathcal{R} + \gamma^h V_k(s_{t+h}) \right] \tag{18}$$

where $\mathcal{R}$ is the discounted cumulative return $\sum_{k=0}^{h} \gamma^k r_{t+k}$ within the chunk. If the goal state is reached at step $i \leq H$, the return is truncated, and no bootstrapping from $V_k$ is performed. In our experiments, we set the chunk horizon $h = 4$.

**3. Chunk Guided Q-Learning (CGQ)** The CGQ backup performs a convex combination of the 1-step and chunk-based updates, weighted by a hyperparameter $w = 0.7$:

$$V_{k+1} \leftarrow w \cdot V_{\text{single}} + (1 - w) \cdot V_{\text{chunk}} \tag{19}$$

**Evaluation.** Gridworld is a tabular environment, where we do not have a source of bias such as function approximation error, as in deep RL. To simulate the accumulation of errors in single-step TD learning, we added Gaussian noise with $\sigma = 0.05$ to each value update. We calculate the accuracy of value function with mean squared error $\mathbb{E}_{s \in D}[((V(s) - V^*(s))^2]$.

## A.7. Q&As

We use `cube-double` environment for Q&A and all ablation studies. For the critic design experiments, we use the same $\alpha$, $\alpha_{\text{step}}$, $\beta$, and $\tau$ in CGQ across all variants. For ablation studies on horizon length, $\beta$, and $\tau$, except for the target hyperparameter, all other hyperparameter values follow those of the `cube-double` setting reported in Table 10.

## B. Hyperparameters

**Shared hyperparameters.** Here, we report the shared hyperparameters across all methods in Table 4.

*Table 4.* **Shared hyperparameters across all methods.**

| Hyperparameter | Value |
| --- | --- |
| Learning rate | 0.0003 |
| Optimizer | Adam (Kingma & Ba, 2015) |
| Gradient steps | 1000000 (default), 2500000 (cube-triple, cube-quadruple) |
| Batch size | 256 (default), 1024 (cube-triple, Cube-quadruple) |

**Network architectures.** For the network size, we consider $[512, 512, 512, 512]$- and $[1024, 1024, 1024, 1024]$-sized MLPs for all the baselines except DEAS, using a larger architecture for `cube-triple` and `cube-quadruple` tasks and the smaller one otherwise. For DEAS, we follow the original setup and use $[512, 512, 512, 512]$ for the policy, $[256, 256, 256, 256]$ for the critic, and $[128, 128, 128, 128]$ for the value function.

**Hyperparameters of CGQ** Our main hyperparameters are BC constraint for the chunked critic $\alpha$, one for the single step critic $\alpha_{\text{step}}$, distillation coefficient $\beta$ and distillation expectile $\tau$. We report those hyperparameters tuning range in Table 6 and hyperparameters for all the tasks in Table 10.

*Table 5.* **Hyperparameters of CGQ.**

| Hyperparameter | Value |
| --- | --- |
| Learning rate | 0.0003 |
| Optimizer | Adam (Kingma & Ba, 2015) |
| Gradient steps | 1000000 (default), 2500000 (cube-triple, cube-quadruple) |
| Batch size | 256 (default), 1024 (cube-triple, Cube-quadruple) |
| Discount factor $\gamma$ | 0.99 |
| Critic Ensemble Size | 2 |
| Clipped Q-Learning | False |
| Flow-steps | 10 |
| Flow time sampling distribution | $\text{Unif}([0, 1])$ |
| Chunk size for $Q_{\phi_c}$ | 10 |
| BC coefficient $\alpha, \alpha_{step}$ | Table 10 |
| Distillation coefficient $\beta$ | Table 10 |
| Distillation expectile $\tau$ | 0.95 (default) |

*Table 6.* **Hyperparameter tuning range for CGQ**.

| Environment | BC Constraint for $\pi_{\theta_c}$ ($\alpha$) | BC Constraint for $\pi_\theta$ ($\alpha_{\text{step}}$) | Distillation Coefficient ($\beta$) | Distillation Expectile ($\tau$) | Chunk size ($h$) |
| --- | --- | --- | --- | --- | --- |
| **Manipulation** | $\{100, 300\}$ | $\{100, 300\}$ | $\{1, 0.1, 0.01\}$ | $\{0.5, 0.8, 0.95\}$ | $\{10, 20\}$ |
| **Navigation** | $\{30, 100\}$ | $\{10, 30\}$ | $\{1, 0.5, 0.1, 0.01\}$ | $\{0.5, 0.6, 0.7, 0.8, 0.95\}$ | $\{10\}$ |

**BC Constraint ($\alpha$).** For QC-FQL and NFQL, we adopt the default hyperparameter $\alpha$ from FQL for each domain, and tune all methods on the default task (`singletask-v0`) using three choices of $\alpha \in \{\alpha_{\text{default}}/3, \alpha_{\text{default}}, 3\alpha_{\text{default}}\}$, following (Park et al., 2025d). For DEAS, we sweep $\alpha$ over $\{0.03, 0.1, 0.3, 1.0, 3.0, 10.0\}$ following the tuning range.

**Chunk Size $h$.** Here, we clarify the chunk sizes used to obtain the results reported in the tables. We follow the chunk size settings used in the original papers for each method. For some baselines (e.g., NFQL and QC-FQL), results are drawn from multiple prior works, which use different chunk sizes across tasks. Accordingly, we report the task-specific chunk size used for each method, including those adopted from prior work. See Table 9.

*Table 7.* **Hyperparameter tuning range for DEAS.**

| BC Constraint $(\alpha)$ | Chunk size $(h)$ | Intra-chunk discount $(\gamma_1)$ | HL-Gaussian support $(s)$ |
|---|---|---|---|
| $\{0.03, 0.1, 0.3, 1, 3, 10\}$ | $\{4, 8\}$ | $\{0.8, 0.9, 0.99, 0.999\}$ | {data-centric, universal} |

*Table 8.* **Hyperparameter tuning range for DQC.**

| Environment | Backup Quantile $(\kappa_b)$ | Distillation Expectile $(\kappa_d)$ | Backup Horizon $(h)$ or $(n)$ | Policy Chunk Size $(h_a)$ |
|---|---|---|---|---|
| `cube-*` | $\{0.5, 0.7, 0.9, 0.93, 0.95, 0.97, 0.99\}$ | $\{0.5, 0.8\}$ | $\{5, 10, 25\}$ | $\{1, 5, 25\}$ |
| **Others** | $\{0.5, 0.7, 0.9\}$ | $\{0.5, 0.8\}$ | $\{5, 10, 25\}$ | $\{1, 5, 25\}$ |

*Table 9.* **Task-specific chunk size configurations.** This table summarizes the chunk sizes used by each agent for each task, including both our experimental settings and those reported in prior work. Gray entries indicate chunk sizes corresponding to results reported in prior work.

| Environment | NFQL | QC-FQL | DEAS | DQC | CGQ |
|---|---|---|---|---|---|
| `antmaze-large` | 5 | 5 | 4 | 25 | 10 |
| `antmaze-giant` | 5 | 5 | 4 | 25 | 10 |
| `humanoidmaze-medium` | 5 | 5 | 4 | 10 | 10 |
| `puzzle-3x3-sparse` | 5 | 5 | 4 | 10 | 20 |
| `scene-sparse` | 5 | 5 | 4 | 5 | 20 |
| `cube-double` | 5 | 5 | 4 | 5 | 20 |
| `cube-triple` | 4 | 4 | 4 | 25 | 20 |
| `puzzle-4x4` | 5 | 5 | 8 | 5 | 10 |
| `cube-quadruple` | 4 | 4 | 4 | 10 | 20 |

**Hyperparameter search for QC-FQL.** Following the tuning process of the original paper (Li et al., 2025b), we set the optimal $\alpha$ of FQL (Park et al., 2025d) as $\alpha_{\text{default}}$ and sweep over $\{\alpha_{\text{default}}/3, \alpha_{\text{default}}, 3\alpha_{\text{default}}\}$. For the puzzle-4x4 and navigation tasks, we use the chunk size 5. See Table 10 for the value of $\alpha$.

**Hyperparameter search for DQC.** We perform an exhaustive hyperparameter sweep for DQC, evaluating 98 configurations for `cube-*` environments and 42 configurations for the others. In particular, we tune the backup quantile $\kappa_b$, distillation expectile $\kappa_d$, backup horizon $h$ and policy chunk size $h_a$. We use clipped Q-learning and use 32 as the number of Best-of-$N$ samples following the original paper. See Table 8 for the full hyperparameter sweep range.

**Hyperparameter search for DEAS.** We perform an extensive hyperparameter sweep for DEAS, evaluating 96 configurations per environment. In particular, we tune the BC constraint $\alpha$, chunk size $h$, and intra-chunk discount $\gamma_1$. For selecting $v_{\min}$ and $v_{\max}$ in the distributional RL component, we evaluate both approaches proposed in the original paper, *data-centric* and *universal*, and select the better-performing protocol. We denote this selection protocol as $s$ in the table, where $d$ and $u$ correspond to data-centric and universal, respectively. We additionally apply Q-loss normalization and clipped Q-learning following the original setting. See Table 7 for the full hyperparameter sweep range.

**Hyperparameter search for NFQL.** We use $h$ to 5 following Li et al. (2025b). Otherwise, we use the same strategy as in QC-FQL.

*Table 10.* **Task-specific hyperparameters for all methods.**

| Environment | CGQ $(\alpha, \alpha_{step}, \beta, \tau, h)$ | DQC $(h, h_a, \kappa_b, \kappa_d)$ | DEAS $(\alpha, h, \gamma_1, \gamma_2, s)$ | NFQL $(\alpha, h)$ | QC-FQL $(\alpha, h)$ |
|---|---|---|---|---|---|
| antmaze-large | $(100, 10, 0.01, 0.7, 10)$ | $(25, 1, 0.5, 0.8)$ | $(1, 4, 0.99, 0.999, d)$ | $(30, 5)$ | $(30, 5)$ |
| antmaze-giant | $(300, 10, 0.01, 0.8, 10)$ | $(25, 1, 0.9, 0.5)$ | $(0.3, 4, 0.99, 0.999, u)$ | $(30, 5)$ | $(10, 5)$ |
| humanoidmaze-medium | $(100, 10, 0.5, 0.8, 10)$ | $(10, 1, 0.9, 0.5)$ | $(1, 4, 0.8, 0.999, d)$ | $(30, 5)$ | $(30, 5)$ |
| puzzle-3x3-sparse | $(100, 100, 0.01, 0.8, 20)$ | $(10, 5, 0.7, 0.8)$ | $(3, 8, 0.9, 0.99)$ | - | - |
| scene-sparse | $(300, 100, 0.1, 0.8, 20)$ | $(5, 1, 0.9, 0.5)$ | $(3, 4, 0.99, 0.999, u)$ | - | - |
| cube-double | $(100, 100, 1, 0.8, 20)$ | $(5, 1, 0.99, 0.8)$ | - | - | - |
| cube-triple | $(300, 300, 0.01, 0.8, 20)$ | $(25, 5, 0.7, 0.8)$ | - | - | - |
| puzzle-4x4 | $(300, 300, 0.1, 0.5, 10)$ | $(5, 1, 0.7, 0.5)$ | $(1, 4, 0.999, 0.999, u)$ | $(1000, 5)$ | $(3000, 5)$ |
| cube-quadruple | $(300, 300, 0.01, 0.95, 20)$ | $(10, 5, 0.9, 0.8)$ | - | - | - |

*Table 11.* **Hyperparameters for additional experiments.** We report the hyperparameters used for the additional experiments in Table 13, including D4RL, OGBench, pixel-based, and offline-to-online settings. For manipulation tasks, we adopt the semi-sparse reward setting unless otherwise specified. For `puzzle-3x3`, `visual-scene-play-task1`, and `visual-puzzle-3x3-task1`, we normalize the Q-loss, which renders a invariant to the scale of Q values.

| Setting | Environment | CGQ $(\alpha, \alpha_{\text{step}}, \beta, \tau, h)$ |
|---|---|---|
| D4RL | antmaze-umaze-v2 | $(90, 10, 0.01, 0.8, 10)$ |
| | antmaze-umaze-diverse-v2 | $(30, 10, 0.01, 0.8, 10)$ |
| | antmaze-medium-play-v2 | $(90, 10, 0.01, 0.8, 10)$ |
| | antmaze-medium-diverse-v2 | $(90, 10, 0.01, 0.8, 10)$ |
| | antmaze-large-play-v2 | $(27, 3, 0.01, 0.8, 10)$ |
| | antmaze-large-diverse-v2 | $(27, 3, 0.01, 0.8, 10)$ |
| | pen-expert-v1 | $(9000, 3000, 1, 0.8, 10)$ |
| | door-expert-v1 | $(270000, 30000, 1, 0.8, 10)$ |
| | hammer-expert-v1 | $(90000, 30000, 1, 0.8, 10)$ |
| | relocate-expert-v1 | $(270000, 30000, 1, 0.8, 10)$ |
| OGBench | puzzle-3x3 | $(30, 1, 0.01, 0.5, 10)$ |
| | antsoccer-arena | $(30, 10, 0.1, 0.95, 10)$ |
| | humanoidmaze-large | $(30, 10, 0.01, 0.95, 10)$ |
| Pixel-based | visual-cube-single-task1 | $(300, 100, 0.1, 0.5, 10)$ |
| | visual-cube-double-task1 | $(300, 100, 1, 0.5, 10)$ |
| | visual-scene-play-task1 | $(10, 1, 1, 0.5, 10)$ |
| | visual-puzzle-3x3-task1 | $(10, 1, 1, 0.8, 10)$ |
| | visual-puzzle-4x4-task1 | $(300, 100, 0.1, 0.8, 10)$ |
| Offline-to-online | humanoidmaze-medium | $(100, 10, 0.5, 0.8, 10)$ |
| | cube-double | $(100, 100, 1, 0.95, 10)$ |
| | puzzle-4x4 | $(300, 300, 0.1, 0.5, 10)$ |

## C. Complete Numerical Results

**Full results.** For completeness, we provide the complete per-task results for OGBench experiments in Table 12 (corresponding to Table 1). The results are averaged over 4 seeds and we report the standard deviations for each tasks. We highlight the numbers that are above or equal to 95% of the best performance.

*Table 12.* **Complete results for OGBench experiments.** We present the full results on 45 tasks. The results are averaged over 4 seeds.

| Task Type | Task Category | FQL | NFQL | QC-FQL | DEAS | DQC | CGQ |
|---|---|---|---|---|---|---|---|
| Manipulation | scene-task1 | 69 | 22 | 99 | $99_{\pm 0}$ | $94_{\pm 4}$ | $100_{\pm 1}$ |
| | scene-task2 | 51 | 4 | 88 | $98_{\pm 3}$ | $93_{\pm 3}$ | $99_{\pm 2}$ |
| | scene-task3 | 68 | 1 | 98 | $84_{\pm 3}$ | $62_{\pm 7}$ | $99_{\pm 2}$ |
| | scene-task4 | 75 | 50 | 92 | $99_{\pm 1}$ | $84_{\pm 4}$ | $73_{\pm 12}$ |
| | scene-task5 | 25 | 14 | 41 | $87_{\pm 5}$ | $73_{\pm 7}$ | $59_{\pm 11}$ |
| | cube-double-task1 | 63 | 33 | 66 | 76 | $51_{\pm 7}$ | $81_{\pm 9}$ |
| | cube-double-task2 | 36 | 8 | 43 | 51 | $28_{\pm 6}$ | $77_{\pm 3}$ |
| | cube-double-task3 | 22 | 1 | 38 | 47 | $22_{\pm 3}$ | $78_{\pm 4}$ |
| | cube-double-task4 | 9 | 0 | 11 | 8 | $4_{\pm 1}$ | $9_{\pm 1}$ |
| | cube-double-task5 | 12 | 13 | 37 | 57 | $48_{\pm 5}$ | $74_{\pm 7}$ |
| | puzzle-3x3-task1 | 99 | 100 | 100 | $100_{\pm 0}$ | $100_{\pm 0}$ | $99_{\pm 1}$ |
| | puzzle-3x3-task2 | 99 | 100 | 86 | $100_{\pm 0}$ | $100_{\pm 0}$ | $100_{\pm 0}$ |
| | puzzle-3x3-task3 | 100 | 100 | 50 | $100_{\pm 1}$ | $100_{\pm 0}$ | $97_{\pm 2}$ |
| | puzzle-3x3-task4 | 100 | 100 | 75 | $97_{\pm 5}$ | $100_{\pm 0}$ | $98_{\pm 1}$ |
| | puzzle-3x3-task5 | 100 | 91 | 4 | $100_{\pm 0}$ | $100_{\pm 0}$ | $98_{\pm 2}$ |
| | puzzle-4x4-task1 | 34 | $35_{\pm 5}$ | $48_{\pm 3}$ | $87_{\pm 6}$ | $13_{\pm 6}$ | $65_{\pm 3}$ |
| | puzzle-4x4-task2 | 16 | $21_{\pm 6}$ | $18_{\pm 2}$ | $9_{\pm 10}$ | $11_{\pm 4}$ | $5_{\pm 1}$ |
| | puzzle-4x4-task3 | 18 | $26_{\pm 3}$ | $39_{\pm 4}$ | $60_{\pm 7}$ | $6_{\pm 1}$ | $51_{\pm 11}$ |
| | puzzle-4x4-task4 | 11 | $20_{\pm 4}$ | $18_{\pm 1}$ | $30_{\pm 6}$ | $4_{\pm 3}$ | $18_{\pm 5}$ |
| | puzzle-4x4-task5 | 7 | $13_{\pm 1}$ | $16_{\pm 3}$ | $10_{\pm 2}$ | $3_{\pm 1}$ | $2_{\pm 0}$ |
| | cube-triple-task1 | 31 | 17 | 100 | 98 | $8_{\pm 3}$ | $100_{\pm 0}$ |
| | cube-triple-task2 | 9 | 91 | 92 | 95 | $0_{\pm 0}$ | $100_{\pm 0}$ |
| | cube-triple-task3 | 12 | 0 | 92 | 88 | $0_{\pm 0}$ | $96_{\pm 0}$ |
| | cube-triple-task4 | 0 | 0 | 59 | 45 | $0_{\pm 0}$ | $80_{\pm 6}$ |
| | cube-triple-task5 | 2 | 0 | 74 | 87 | $0_{\pm 0}$ | $85_{\pm 9}$ |
| | cube-quadruple-task1 | 79 | 70 | 79 | 92 | $90_{\pm 14}$ | $100_{\pm 0}$ |
| | cube-quadruple-task2 | 0 | 97 | 63 | 100 | $62_{\pm 29}$ | $100_{\pm 0}$ |
| | cube-quadruple-task3 | 6 | 1 | 33 | 62 | $53_{\pm 31}$ | $89_{\pm 9}$ |
| | cube-quadruple-task4 | 0 | 13 | 38 | 31 | $34_{\pm 17}$ | $72_{\pm 3}$ |
| | cube-quadruple-task5 | 0 | 0 | 12 | 35 | $40_{\pm 32}$ | $24_{\pm 12}$ |
| Navigation | antmaze-large-task1 | 80 | $38_{\pm 15}$ | $20_{\pm 9}$ | $74_{\pm 3}$ | $72_{\pm 6}$ | $88_{\pm 2}$ |
| | antmaze-large-task2 | 57 | $39_{\pm 2}$ | $0_{\pm 0}$ | $46_{\pm 9}$ | $61_{\pm 7}$ | $74_{\pm 6}$ |
| | antmaze-large-task3 | 93 | $81_{\pm 5}$ | $49_{\pm 8}$ | $80_{\pm 4}$ | $74_{\pm 4}$ | $92_{\pm 3}$ |
| | antmaze-large-task4 | 80 | $38_{\pm 11}$ | $6_{\pm 6}$ | $54_{\pm 9}$ | $73_{\pm 16}$ | $26_{\pm 20}$ |
| | antmaze-large-task5 | 83 | $38_{\pm 22}$ | $23_{\pm 15}$ | $82_{\pm 3}$ | $74_{\pm 6}$ | $55_{\pm 33}$ |
| | antmaze-giant-task1 | 4 | $0_{\pm 0}$ | $0_{\pm 0}$ | $0_{\pm 0}$ | $0_{\pm 1}$ | $0_{\pm 0}$ |
| | antmaze-giant-task2 | 9 | $1_{\pm 1}$ | $0_{\pm 0}$ | $28_{\pm 3}$ | $41_{\pm 4}$ | $0_{\pm 1}$ |
| | antmaze-giant-task3 | 0 | $0_{\pm 0}$ | $0_{\pm 0}$ | $1_{\pm 1}$ | $1_{\pm 1}$ | $0_{\pm 0}$ |
| | antmaze-giant-task4 | 14 | $0_{\pm 0}$ | $0_{\pm 0}$ | $8_{\pm 4}$ | $5_{\pm 2}$ | $0_{\pm 0}$ |
| | antmaze-giant-task5 | 16 | $8_{\pm 3}$ | $0_{\pm 0}$ | $2_{\pm 3}$ | $2_{\pm 1}$ | $17_{\pm 8}$ |
| | humanoidmaze-medium-task1 | 19 | $3_{\pm 3}$ | $0_{\pm 0}$ | $22_{\pm 4}$ | $87_{\pm 4}$ | $0_{\pm 0}$ |
| | humanoidmaze-medium-task2 | 94 | $26_{\pm 11}$ | $1_{\pm 1}$ | $41_{\pm 6}$ | $94_{\pm 3}$ | $80_{\pm 3}$ |
| | humanoidmaze-medium-task3 | 74 | $49_{\pm 8}$ | $0_{\pm 0}$ | $59_{\pm 7}$ | $94_{\pm 2}$ | $26_{\pm 48}$ |
| | humanoidmaze-medium-task4 | 3 | $4_{\pm 1}$ | $0_{\pm 0}$ | $7_{\pm 2}$ | $90_{\pm 3}$ | $0_{\pm 1}$ |
| | humanoidmaze-medium-task5 | 97 | $32_{\pm 3}$ | $17_{\pm 5}$ | $56_{\pm 6}$ | $98_{\pm 0}$ | $62_{\pm 26}$ |

**Additional task coverage.** We further evaluate CGQ on a broader set of tasks to examine its applicability across diverse settings, as shown in Table 13. Our evaluation includes standard D4RL environments (Fu et al., 2020), widely used benchmarks for offline RL, additional state-based and pixel-based OGBench tasks, and offline-to-online fine-tuning.

*Table 13.* **Additional experimental results.** The results are averaged over 4 seeds unless otherwise specified.

| Setting | Task | CGQ | FQL |
|---|---|---|---|
| D4RL | antmaze-umaze-v2 | $95_{\pm 1}$ | 96 |
| | antmaze-umaze-diverse-v2 | $78_{\pm 7}$ | 89 |
| | antmaze-medium-play-v2 | $75_{\pm 5}$ | 78 |
| | antmaze-medium-diverse-v2 | $75_{\pm 4}$ | 71 |
| | antmaze-large-play-v2 | $79_{\pm 10}$ | 84 |
| | antmaze-large-diverse-v2 | $81_{\pm 5}$ | 83 |
| | pen-expert-v1 | $143_{\pm 2}$ | 142 |
| | door-expert-v1 | $104_{\pm 0}$ | 104 |
| | hammer-expert-v1 | $122_{\pm 2}$ | 125 |
| | relocate-expert-v1 | $108_{\pm 1}$ | 107 |
| OGBench | puzzle-3x3-semi-sparse | $85_{\pm 6}$ | 32 |
| | antsoccer-arena | $63_{\pm 2}$ | 60 |
| | humanoidmaze-large | $1_{\pm 0}$ | 4 |
| Pixel-based | visual-cube-single-task1 | $49_{\pm 12}$ | $25_{\pm 7}$ |
| | visual-cube-double-task1 | $19_{\pm 7}$ | $1_{\pm 7}$ |
| | visual-scene-play-task1 | $93_{\pm 1}$ | $88_{\pm 4}$ |
| | visual-puzzle-3x3-task1 | $47_{\pm 14}$ | $33_{\pm 5}$ |
| | visual-puzzle-4x4-task1 | $21_{\pm 2}$ | $10_{\pm 3}$ |
| Offline-to-online | humanoidmaze-medium | $7_{\pm 7} \rightarrow 34_{\pm 16}$ | $12 \rightarrow 22$ |
| | cube-double | $40_{\pm 3} \rightarrow 98_{\pm 2}$ | $40 \rightarrow 92$ |
| | puzzle-4x4 | $17_{\pm 5} \rightarrow 100_{\pm 0}$ | $8 \rightarrow 38$ |

**Training curves.** We provide the training curve of CGQ for experiments in OGBench manipulation and locomotion environments in Figure 7 and Figure 8 (corresponding to Table 1). We plot the mean and the standard deviation (across 4 seeds) by covering [mean - std, mean + std] area with a lighter color.

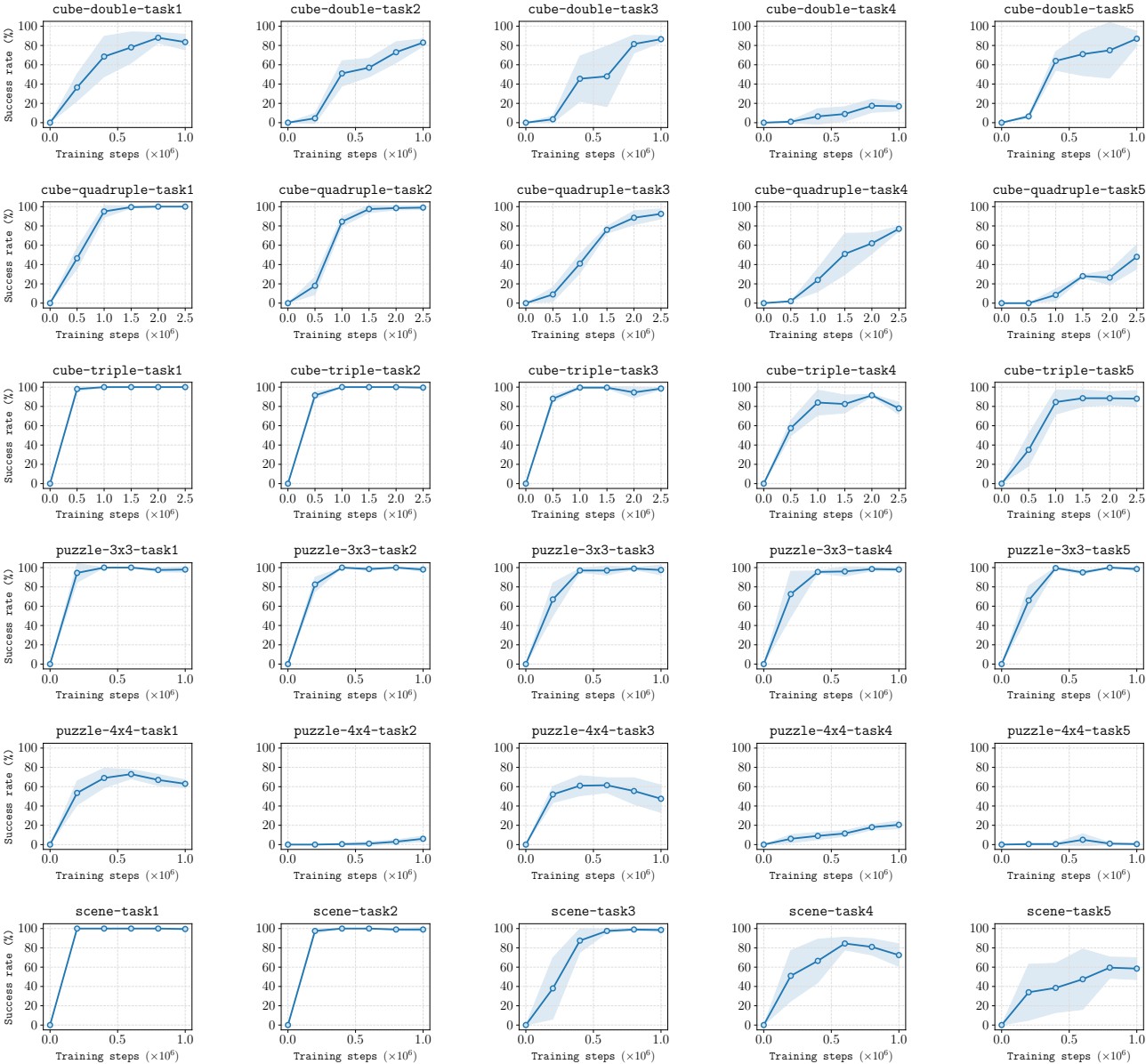

*Figure 7.* **Training curve of CGQ in OGBench manipulation environments.** We report the success rate for 50 evaluation episodes across 4 seeds (total 200 episodes). Shaded region represents the [mean - std, mean + std].

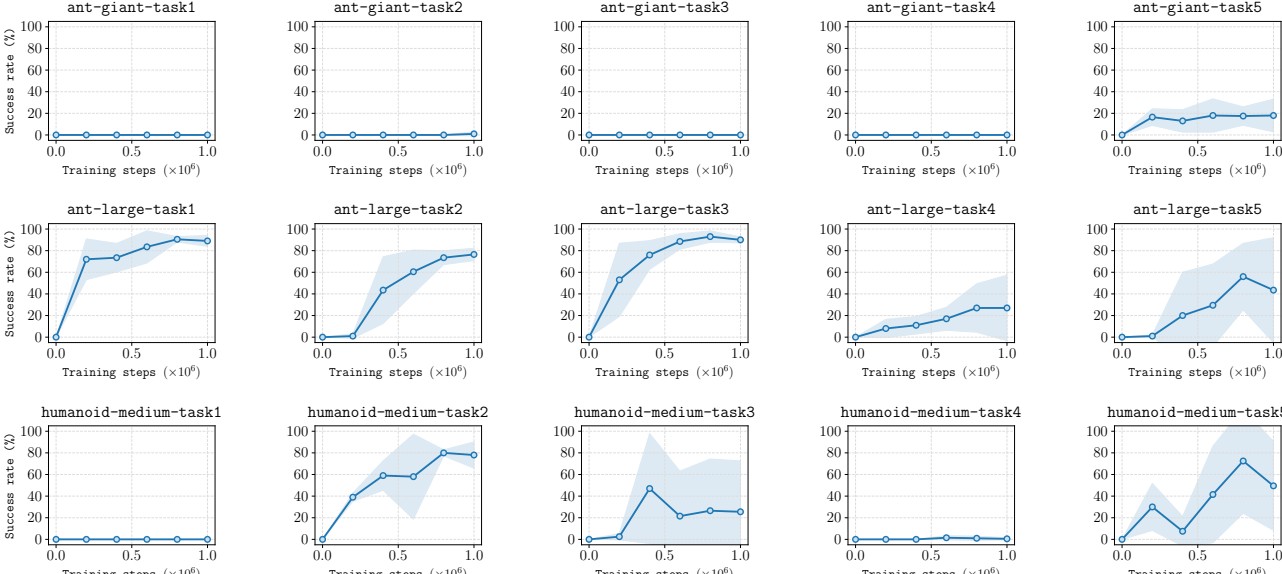

*Figure 8.* **Training curve of CGQ in OGBench locomotion environments.** We report the success rate for 50 evaluation episodes across 4 seeds (total 200 episodes). Shaded region represents the [mean - std, mean + std].

# D. Chunk-Guidance Reliability

One potential limitation of CGQ is that it uniformly uses action chunks from the offline dataset for the regularization in equation 11. When the chunked critic provides poor estimates for some chunks, the resulting guidance can negatively affect the single-step critic. We attribute the weaker gains observed in navigation tasks partly to this unreliable chunk guidance. To investigate whether this issue can be alleviated, we evaluate two simple filtering strategies that exclude potentially unreliable chunks from the guidance. Specifically, we consider Q-value filtering, which removes chunks whose chunked value substantially differs from the corresponding single-step value, and Q-loss filtering, which removes chunks with large chunked TD error. As shown in Figure 9, both filters substantially improve performance on antmaze-large, where unreliable chunk guidance is more prevalent, but have little effect on scene.

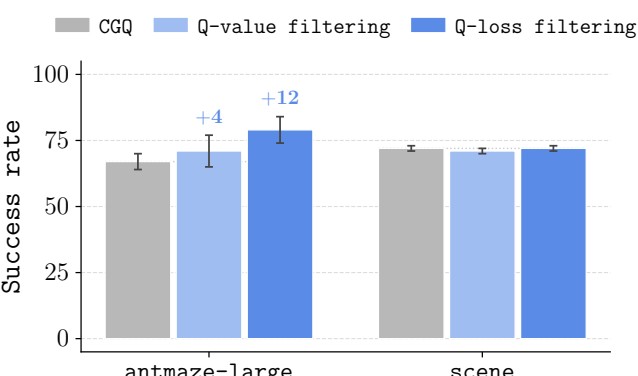

*Figure 9.* **Effect of filtering unreliable chunk guidance.** The results are averaged over 5 tasks per domain and 4 seeds.

For Q-value filtering, we compute the relative gap between the chunked value and the corresponding single-step value:

$$d(s_t, \mathbf{a}_t) = \frac{Q_c(s_t, \mathbf{a}_t) - Q(s_t, a_t)}{|Q(s_t, a_t)|}, \tag{20}$$

where $\mathbf{a}_t = (a_t, \ldots, a_{t+h-1})$ denotes the action chunk. We filter out action chunks whose ratio exceeds a threshold, treating large positive gaps as potentially overestimated and unreliable for guidance. We sweep the threshold over $\{0.05, 0.1, 0.2, 0.3, 0.5, 0.6\}$.

For Q-loss filtering, we compute the per-chunk TD error:

$$\ell(s_t, \mathbf{a}_t) = \left( Q_c(s_t, \mathbf{a}_t) - \sum_{i=0}^{h-1} \gamma^i r_{t+i} - \gamma^h \bar{Q}_c(s_{t+h}, \mathbf{a}_{t+h}) \right)^2. \tag{21}$$

We filter out action chunks above the $\rho$-th quantile of this loss, treating poorly fitted chunks as unreliable for guidance. We sweep $\rho$ over $\{0.2, 0.4, 0.6, 0.8, 0.9, 0.99\}$.

# E. Proofs

Here, we prove the main theorems of the paper. Throughout the proof, we consider the Hilbert space $\mathcal{Q} = L_2(\mathcal{S} \times \mathcal{A}, \|\cdot\|)$ of state-action value functions, equipped with the weighted norm $\|Q\| = (\mathbb{E}_{(s,a)\sim\mu}[Q(s,a)^2])^{1/2}$, where $\mu$ denotes the state-action distribution of an offline dataset $\mathcal{D}$.

## E.1. Proof of Theorem 4.1

**Theorem E.1** (**Bias accumulation of one-step TD learning**). *Let the operator $\mathcal{T}$ be $\gamma$-Lipschitz operator, i.e., $\|\mathcal{T}Q_1 - \mathcal{T}Q_2\| \leq \gamma\|Q_1 - Q_2\|$, with $\gamma < 1$. Consider a stochastic iterative process $Q_{k+1} = \widehat{\mathcal{T}}Q_k$, where*

$$\widehat{\mathcal{T}}Q_k = \mathcal{T}Q_k + \varepsilon_k, \tag{22}$$

*which $\varepsilon_k$ is a sequence of independent, zero-mean noise vectors with $\mathbb{E}\left[\|\varepsilon_k\|^2\right] \leq \sigma^2$. Let $Q^*$ be the unique fixed point of $\mathcal{T}$ (i.e., $Q^* = \mathcal{T}Q^*$). Then, the asymptotic expected squared error satisfies:*

$$\limsup_{k\to\infty} \mathbb{E}\left[\|Q_k - Q^*\|^2\right] \leq \frac{\sigma^2}{1 - \gamma^2} \tag{23}$$

*Proof.* By substituting the update rule and using the definition of $Q^*$ ($Q^* = \mathcal{T}Q^*$),

$$Q_{k+1} - Q^* = (\mathcal{T}Q_k + \varepsilon_k) - \mathcal{T}Q^* = (\mathcal{T}Q_k - \mathcal{T}Q^*) + \varepsilon_k. \tag{24}$$

Taking the expectation of squared $L_2$ norms:

$$\mathbb{E}[\|Q_{k+1} - Q^*\|^2] = \mathbb{E}\left[\|(\mathcal{T}Q_k - \mathcal{T}Q^*) + \varepsilon_k\|^2\right], \tag{25}$$

which can be expanded as

$$\mathbb{E}[\|Q_{k+1} - Q^*\|^2] = \mathbb{E}[\|\mathcal{T}Q_k - \mathcal{T}Q^*\|^2] + \mathbb{E}[\|\varepsilon_k\|^2] + 2\mathbb{E}\left[\langle\mathcal{T}Q_k - \mathcal{T}Q^*, \varepsilon_k\rangle\right]. \tag{26}$$

Since $\varepsilon_k$ have zero mean and indepedent from $Q_k$, we can remove the cross-term:

$$\mathbb{E}[\|Q_{k+1} - Q^*\|^2] = \mathbb{E}[\|\mathcal{T}Q_k - \mathcal{T}Q^*\|^2] + \mathbb{E}[\|\varepsilon_k\|^2]. \tag{27}$$

Since $\mathbb{E}\left[\|\varepsilon\|^2\right] \leq \sigma^2$,

$$\mathbb{E}[\|Q_{k+1} - Q^*\|^2] \leq \mathbb{E}[\|\mathcal{T}Q_k - \mathcal{T}Q^*\|^2] + \sigma^2. \tag{28}$$

and by the Lipschitz property of $\mathcal{T}$,

$$\mathbb{E}[\|Q_{k+1} - Q^*\|^2] \leq \gamma^2\mathbb{E}[\|Q_k - Q^*\|^2] + \sigma^2. \tag{29}$$

By rolling out the inequality, we have

$$\mathbb{E}[\|Q_{k+1} - Q^*\|^2] \leq \gamma^{2k+2}\mathbb{E}[\|Q_0 - Q^*\|^2] + \sum_{i=0}^{k}\gamma^{2i}\sigma^2. \tag{30}$$

Applying the limit $k \to \infty$ to each side of the equation concludes the proof:

$$\lim_{k\to\infty} \mathbb{E}\left[\|Q_k - Q^*\|^2\right] \leq \frac{\sigma^2}{1 - \gamma^2} \tag{31}$$

$\square$

On top of that, with the assumption that the curvature of $\mathcal{T}$ is bounded, we can derive the bound for the difference between the expectation of the result of the stochastic process and $Q^*$.

**Lemma E.2.** *Consider a stochastic iterative process $Q_{k+1} = \widehat{\mathcal{T}}Q_k$, where*

$$\widehat{\mathcal{T}}Q_k = \mathcal{T}Q_k + \epsilon_k, \tag{32}$$

*and $Q^*$ be the fixed point of $\mathcal{T}$. Suppose that $\mathcal{T}$ has a bounded curvature; i.e.,*

$$\|\mathcal{T}(Q_1) - \mathcal{T}(Q_2) - D\mathcal{T}(Q_2)(Q_1 - Q_2)\| \leq \frac{L}{2}\|Q_1 - Q_2\|^2, \tag{33}$$

*where $D\mathcal{T}$ is a Fréchet derivative of $\mathcal{T}$. Then, the difference between the expectation of the process and the $Q^*$ is bounded as:*

$$\limsup_{k \to \infty} \|\mathbb{E}[Q_k] - Q^*\| \leq \frac{L}{2} \cdot \frac{\sigma^2}{(1 - \gamma^2)(1 - \gamma)}, \tag{34}$$

*Proof.* Since $\epsilon_k$ has zero mean,

$$\mathbb{E}[Q_{k+1}] - Q^* = \mathbb{E}[\mathcal{T}Q_k] - Q^*. \tag{35}$$

Since $Q^*$ is not a random variable,

$$\mathbb{E}[Q_{k+1}] - Q^* = \mathbb{E}[\mathcal{T}Q_k - Q^*]. \tag{36}$$

By the first-order taylor expansion of $\mathcal{T}$ in $Q^*$: $\mathcal{T}(Q) = \mathcal{T}(Q^*) + D\mathcal{T}(Q^*)(Q - Q^*) + R(Q - Q^*)$,

$$\mathbb{E}[Q_{k+1}] - Q^* = \mathbb{E}[D\mathcal{T}(Q^*)(Q_k - Q^*) + R(Q_k - Q^*)]. \tag{37}$$

Split the expectation and taking the norm,

$$\|\mathbb{E}[Q_{k+1}] - Q^*\| = \|\mathbb{E}[D\mathcal{T}(Q^*)(Q_k - Q^*)] + \mathbb{E}[R(Q_k - Q^*)]\|. \tag{38}$$

By triangle inequality,

$$\|\mathbb{E}[Q_{k+1}] - Q^*\| \leq \|\mathbb{E}[D\mathcal{T}(Q^*)(Q_k - Q^*)]\| + \|\mathbb{E}[R(Q_k - Q^*)]\|. \tag{39}$$

Since $D\mathcal{T}(Q^*)$ is a linear operator,

$$\|\mathbb{E}[Q_{k+1}] - Q^*\| \leq \|D\mathcal{T}(Q^*)(\mathbb{E}[Q_k] - Q^*)\| + \|\mathbb{E}[R(Q_k - Q^*)]\|. \tag{40}$$

Since $\mathcal{T}$ is a $\gamma$-Lipschitz operator, $\|D\mathcal{T}(Q^*)\| \leq \gamma$, thus

$$\|\mathbb{E}[Q_{k+1}] - Q^*\| \leq \gamma\|\mathbb{E}[Q_k] - Q^*\| + \|\mathbb{E}[R(Q_k - Q^*)]\|. \tag{41}$$

As $\mathcal{T}$ has a curvature bounded by $L$, $\|R(Q_k - Q^*)\| \leq \frac{L}{2}\|Q_k - Q^*\|^2$. Thus,

$$\|\mathbb{E}[Q_{k+1}] - Q^*\| \leq \gamma\|\mathbb{E}[Q_k] - Q^*\| + \frac{L}{2}\mathbb{E}[\|Q_k - Q^*\|^2]. \tag{42}$$

Taking the limit:

$$\limsup_{k \to \infty} \|\mathbb{E}[Q_{k+1}] - Q^*\| \leq \limsup_{k \to \infty} \gamma\|\mathbb{E}[Q_k] - Q^*\| + \limsup_{k \to \infty} \frac{L}{2}\mathbb{E}[\|Q_k - Q^*\|^2]. \tag{43}$$

By Theorem E.1, we have

$$\limsup_{k \to \infty} \|\mathbb{E}[Q_{k+1}] - Q^*\| \leq \limsup_{k \to \infty} \gamma\|\mathbb{E}[Q_k] - Q^*\| + \frac{L}{2} \cdot \frac{\sigma^2}{1 - \gamma^2}. \tag{44}$$

Since $k \to \infty$, we can have

$$\limsup_{k \to \infty} (1 - \gamma) \| \mathbb{E}\left[Q_k\right] - Q^* \| \leq \frac{L}{2} \cdot \frac{\sigma^2}{1 - \gamma^2}, \tag{45}$$

which concludes the proof:

$$\limsup_{k \to \infty} \| \mathbb{E}\left[Q_k\right] - Q^* \| \leq \frac{L}{2} \cdot \frac{\sigma^2}{(1 - \gamma^2)(1 - \gamma)}, \tag{46}$$

$\square$

### E.2. Proof of Theorem 4.2

**Lemma E.3.** *Let $\mathcal{F}$ be a normed vector space and let $\mathcal{T} : \mathcal{F} \to \mathcal{F}$ be $\gamma$-Lipschitz operator in the $L_2$ norm, i.e., $\|\mathcal{T}Q_1 - \mathcal{T}Q_2\| \leq \gamma \|Q_1 - Q_2\|$, with $\gamma < 1$. Let its fixed point of $\mathcal{T}$ be $x^*$ ($\mathcal{T}x^* = x^*$). Then for any $x \in \mathcal{F}$, we have the following inequality:*

$$\|x - x^*\| \leq \frac{1}{1 - \gamma} \|\mathcal{T}x - x\|. \tag{47}$$

*Proof.* We write

$$x - x^* = x - \mathcal{T}x + \mathcal{T}x - \mathcal{T}x^*. \tag{48}$$

Taking norms and using the triangle inequality and the linearlity of $\mathcal{T}$,

$$\|x - x^*\| \leq \|x - \mathcal{T}x\| + \|\mathcal{T}x - \mathcal{T}x^*\|. \tag{49}$$

Since $T$ is a contraction with factor $\gamma$, $\|\mathcal{T}x - \mathcal{T}x^*\| \leq \gamma \|x - x^*\|$ holds, thus

$$\|x - x^*\| \leq \|x - \mathcal{T}x\| + \gamma \|x - x^*\|. \tag{50}$$

By rearranginging the inequality,

$$(1 - \gamma)\|x - x^*\| \leq \|x - \mathcal{T}x\|, \tag{51}$$

which proves the theorem:

$$\|x - x^*\| \leq \frac{1}{1 - \gamma} \|\mathcal{T}x - x\|. \tag{52}$$

$\square$

**Lemma E.4.** *Let $\mathcal{T}_\beta$ be the regularized operator for some $Q_c \in \mathcal{Q}$: $\mathcal{T}_\beta Q = \frac{1}{1+\beta} \mathcal{T}Q + \frac{\beta}{1+\beta} Q_c$. Then $\mathcal{T}_\beta$ is Lipschitz with factor of $\gamma/(1+\beta)$, i.e.,*

$$\|\mathcal{T}_\beta Q_1 - \mathcal{T}_\beta Q_2\| \leq \frac{\gamma}{1+\beta} \|Q_1 - Q_2\|. \tag{53}$$

*Proof.* By definition,

$$\|\mathcal{T}_\beta Q_1 - \mathcal{T}_\beta Q_2\| = \frac{1}{1+\beta} \mathcal{T}Q_1 - \frac{1}{1+\beta} \mathcal{T}Q_2. \tag{54}$$

Since $\mathcal{T}$ is a contraction operator with factor $\gamma$,

$$\|\mathcal{T}_\beta Q_1 - \mathcal{T}_\beta Q_2\| \leq \frac{\gamma}{1+\beta} \|Q_1 - Q_2\|, \tag{55}$$

which concludes the proof. $\square$

**Lemma E.5.** *Let $Q_\beta^*$ be the fixed point of $\mathcal{T}_\beta$. Then*

$$\|Q^* - Q_\beta^*\| \leq \frac{\beta}{1 - \gamma + \beta}\|Q^* - Q_c\|. \tag{56}$$

*Proof.* From Lemma E.3 and Lemma E.4, we have

$$\|Q^* - Q_\beta^*\| \leq \frac{1}{1 - \gamma/(1 + \beta)}\|\mathcal{T}_\beta Q^* - Q^*\|. \tag{57}$$

By definition of $\mathcal{T}_\beta$, we have:

$$\|Q^* - Q_\beta^*\| \leq \frac{1}{1 - \gamma/(1 + \beta)}\|Q^* - \frac{1}{\beta + 1}\mathcal{T}Q^* - \frac{\beta}{\beta + 1}Q_c)\|. \tag{58}$$

Since $\mathcal{T}Q^* = Q^*$,

$$\|Q^* - Q_\beta^*\| \leq \frac{1}{1 - \gamma/(1 + \beta)}\|\frac{\beta}{\beta + 1}(Q^* - Q_c)\|, \tag{59}$$

concluding the proof:

$$\|Q^* - Q_\beta^*\| \leq \frac{\beta}{1 + \beta - \gamma}\|Q^* - Q_c\|. \tag{60}$$

$\square$

**Lemma E.6.** *Consider a stochastic iterative process $\widehat{Q}_\beta^{k+1} = \widehat{\mathcal{T}}_\beta \widehat{Q}_\beta^k$, where*

$$\widehat{\mathcal{T}}_\beta \widehat{Q}_\beta^k = \frac{1}{1 + \beta}\widehat{\mathcal{T}}\widehat{Q}_\beta^k + \frac{\beta}{1 + \beta}Q_c, \tag{61}$$

*Suppose that $\mathcal{T}$ has a curvature bounded by $L$. Then, the difference between the asymptotic expectation of the process and the $Q_\beta^*$ satisfies:*

$$\limsup_{k \to \infty} \|\mathbb{E}\left[\widehat{Q}_\beta^k\right] - Q_\beta^*\| \leq \frac{L}{2} \cdot \frac{(1 + \beta)\sigma^2}{((1 + \beta)^2 - \gamma^2)(1 + \beta - \gamma)}, \tag{62}$$

*Proof.* We can repeat the proof of Lemma E.2 with contraction factor of $\gamma/(1 + \beta)$ and noise level of $\sigma/(1 + \beta)$, leading to the bound:

$$\limsup_{k \to \infty} \|\mathbb{E}\left[\widehat{Q}_\beta^k\right] - Q_\beta^*\| \leq \frac{L}{2} \cdot \frac{(1 + \beta)\sigma^2}{((1 + \beta)^2 - \gamma^2)(1 + \beta - \gamma)}. \tag{63}$$

$\square$

By combining two results, we show the main theorem of the paper:

**Theorem E.7 (Improved bound via CGQ regularization).** *Let $Q^*$ be the fixed point of $\mathcal{T}$, and $\widehat{\mathcal{T}}_\beta$ be a stochastic iterative process $\widehat{\mathcal{T}}_\beta Q = \frac{1}{1+\beta}\widehat{\mathcal{T}}Q + \frac{\beta}{1+\beta}Q_c$. Then, the asymptotic expected squared error satisfies:*

$$\limsup_{k \to \infty} \mathbb{E}\left[\|\widehat{Q}_\beta^k - Q^*\|^2\right] \leq \frac{\sigma^2}{(1 + \beta)^2 - \gamma^2} + \frac{\beta^2\|Q^* - Q_c\|^2}{(1 + \beta - \gamma)^2} + \frac{L(1 + \beta)\beta\|Q^* - Q_c\|\sigma^2}{((1 + \beta)^2 - \gamma^2)(1 + \beta - \gamma)^2}. \tag{64}$$

*Proof.* We can write

$$\limsup_{k \to \infty} \mathbb{E}\left[\|\widehat{Q}_\beta^k - Q^*\|^2\right] = \limsup_{k \to \infty} \mathbb{E}\left[\|(\widehat{Q}_\beta^k - Q_\beta^*) + (Q_\beta^* - Q^*)\|^2\right]. \tag{65}$$

Expanding the formula gives

$$\limsup_{k\to\infty} \mathbb{E}\left[\|\widehat{Q}_\beta^k - Q^*\|^2\right] = \limsup_{k\to\infty} \mathbb{E}\left[\|\widehat{Q}_\beta^k - Q_\beta^*\|^2 + \|Q_\beta^* - Q^*\|^2 + 2\langle\widehat{Q}_\beta^k - Q_\beta^*, Q_\beta^* - Q^*\rangle\right]. \tag{66}$$

Since $\|Q_\beta^* - Q^*\|^2$ is not a random variable,

$$\limsup_{k\to\infty} \mathbb{E}\left[\|\widehat{Q}_\beta^k - Q^*\|^2\right] = \|Q_\beta^* - Q^*\|^2 + \limsup_{k\to\infty} \mathbb{E}\left[\|\widehat{Q}_\beta^k - Q_\beta^*\|^2\right] + 2\langle\mathbb{E}\left[\widehat{Q}_\beta^k - Q_\beta^*\right], Q_\beta^* - Q^*\rangle. \tag{67}$$

By Cauchy-Schwarz inequality,

$$\limsup_{k\to\infty} \mathbb{E}\left[\|\widehat{Q}_\beta^k - Q^*\|^2\right] \leq \|Q_\beta^* - Q^*\|^2 + \limsup_{k\to\infty} \mathbb{E}\left[\|\widehat{Q}_\beta^k - Q_\beta^*\|^2\right] + 2\|\mathbb{E}\left[\widehat{Q}_\beta^k\right] - Q_\beta^*\|\|Q_\beta^* - Q^*\|. \tag{68}$$

From Lemma E.5, Lemma E.6, Lemma E.2, we have

$$\limsup_{k\to\infty} \mathbb{E}\left[\|\widehat{Q}_\beta^k - Q^*\|^2\right] \leq \frac{\sigma^2}{(1+\beta)^2 - \gamma^2} + \frac{\beta^2\|Q^* - Q_c\|^2}{(1+\beta-\gamma)^2} + \frac{L(1+\beta)\beta\|Q^* - Q_c\|\sigma^2}{((1+\beta)^2 - \gamma^2)(1+\beta-\gamma)^2} \tag{69}$$

concluding the proof. $\square$

Here, we are interested in when this bound can be strictly better than pure single-step TD learning ($\beta = 0$) or action-chunked TD learning ($\beta = \infty$). We can derive the condition for this with simple algebra.

**Lemma E.8.** *Let $f(\beta)$ be the bound derived from Theorem E.7,*

$$f(\beta) = \frac{\sigma^2}{(1+\beta)^2 - \gamma^2} + \frac{\beta^2\|Q^* - Q_c\|^2}{(1+\beta-\gamma)^2} + \frac{L(1+\beta)\beta\|Q^* - Q_c\|\sigma^2}{((1+\beta)^2 - \gamma^2)(1+\beta-\gamma)^2}. \tag{70}$$

*Then $\lim_{\beta\to\infty} f'(\beta) = 0^+$.*

*Proof.* We analyze the dominating term for each terms separately.

**Term 1:**

$$f_1(\beta) = \frac{\sigma^2}{(1+\beta)^2 - \gamma^2}. \tag{71}$$

The derivative is

$$f_1'(\beta) = -\frac{2\sigma^2(1+\beta)}{((1+\beta)^2 - \gamma^2)^2}. \tag{72}$$

thus

$$f_1'(\beta) \simeq -\frac{2\sigma^2}{\beta^3}. \tag{73}$$

**Term 2:**

$$f_2(\beta) = \frac{\beta^2\|Q^* - Q_c\|^2}{(1+\beta-\gamma)^2}. \tag{74}$$

Its derivative is

$$f_2'(\beta) = \frac{2\beta\|Q^* - Q_c\|^2(1-\gamma)}{(1+\beta-\gamma)^3}, \tag{75}$$

thus

$$f_2'(\beta) \simeq \frac{2(1-\gamma)\|Q^* - Q_c\|^2}{\beta^2}. \tag{76}$$

**Term 3:**

$$f_3(\beta) = \frac{L(1+\beta)\beta\|Q^* - Q_c\|\sigma^2}{((1+\beta)^2 - \gamma^2)(1+\beta-\gamma)^2}. \tag{77}$$

At $\beta = 0$, the numerator is zero, and its derivative is $(1 + 2\beta)\|Q^* - Q_c\|\sigma^2$, and the denominator evaluates asymptotically to $\beta^4$. Therefore by the quotient rule,

$$f_3'(\beta) \simeq \frac{L\|Q^* - Q_c\|\sigma^2(2\beta \cdot \beta^4 - 4\beta^3 \cdot \beta^2)}{(\beta^4)^2} = \frac{L\|Q^* - Q_c\|\sigma^2}{2\beta^3}. \tag{78}$$

Thus, the limit is dominated by the $1/\beta^2$ term, proving

$$\lim_{\beta \to \infty} f'(\beta) = 0^+. \tag{79}$$

$\square$

**Corollary E.9.** *Let $f(\beta)$ be the bound derived from Theorem E.7,*

$$f(\beta) = \frac{\sigma^2}{(1+\beta)^2 - \gamma^2} + \frac{\beta^2\|Q^* - Q_c\|^2}{(1+\beta-\gamma)^2} + \frac{L(1+\beta)\beta\|Q^* - Q_c\|\sigma^2}{((1+\beta)^2 - \gamma^2)(1+\beta-\gamma)^2}. \tag{80}$$

*When $L\|Q_c - Q^*\| < 2\frac{1-\gamma}{1+\gamma}$, $f(\beta)$ minimized at some $0 < \beta^* < \infty$, and is strictly smaller than both $\frac{\sigma^2}{1-\gamma^2}$ and $\|Q^* - Q_c\|^2$.*

*Proof.* We show that $f'(0) < 0$. We analyze each term separately.

**Term 1:**

$$f_1(\beta) = \frac{\sigma^2}{(1+\beta)^2 - \gamma^2}. \tag{81}$$

The derivative is

$$f_1'(\beta) = -\frac{2\sigma^2(1+\beta)}{((1+\beta)^2 - \gamma^2)^2}. \tag{82}$$

thus

$$f_1'(0) = -\frac{2\sigma^2}{(1-\gamma^2)^2}. \tag{83}$$

**Term 2:**

$$f_2(\beta) = \frac{\beta^2\|Q^* - Q_c\|^2}{(1+\beta-\gamma)^2}. \tag{84}$$

Its derivative is

$$f_2'(\beta) = \frac{2\beta\|Q^* - Q_c\|^2(1-\gamma)}{(1+\beta-\gamma)^3}, \tag{85}$$

thus

$$f_2'(0) = 0. \tag{86}$$

**Term 3:**

$$f_3(\beta) = \frac{L(1+\beta)\beta\|Q^* - Q_c\|\sigma^2}{((1+\beta)^2 - \gamma^2)(1+\beta-\gamma)^2}. \tag{87}$$

At $\beta = 0$, the numerator is zero, and its derivative is $L(1 + 2\beta)\|Q^* - Q_c\|\sigma^2 = L\|Q^* - Q_c\|\sigma^2$, and the denominator evaluates to $(1 - \gamma^2)(1 - \gamma)^2$. Therefore by the quotient rule,

$$f_3'(0) = \frac{L\|Q^* - Q_c\|\sigma^2}{(1 - \gamma^2)(1 - \gamma)^2}. \tag{88}$$

Thus,

$$f'(0) = f_1'(0) + f_2'(0) + f_3'(0) = -\frac{2\sigma^2}{(1 - \gamma^2)^2} + 0 + \frac{L\|Q^* - Q_c\|\sigma^2}{(1 - \gamma)^2(1 - \gamma^2)}. \tag{89}$$

If $L\|Q^* - Q_c\| < \frac{2(1-\gamma)}{(1+\gamma)}$, then

$$f'(0) < -\frac{2\sigma^2}{(1 - \gamma^2)^2} + \frac{2(1 - \gamma)/(1 + \gamma)\sigma^2}{(1 - \gamma)^2(1 - \gamma)^2} = 0. \tag{90}$$

Since $f'$ is a continuous function and $\lim_{\beta \to \infty} f'(\beta) = 0^+$ and $f'(0) < 0$, we have $f'(\beta^*) = 0$ in $0 < \beta^* < \infty$. Moreover, among those $\beta^*$, we have at least one $\beta^*$ that is strictly smaller than $f'(\beta^*)$, as $f'(0) < 0$ and $f'(\infty) = 0^+$.

$\square$

