# OpenReview forum: "Chunk-Guided Q-Learning"
_ICML.cc/2026/Conference — ICML 2026 regular_

### Official Review · Reviewer_Xo5v · 2026-02-16

**Soundness:** 4
**Presentation:** 4
**Significance:** 3
**Originality:** 3
**Overall Recommendation:** 5
**Confidence:** 4

**Summary:**

The authors introduce CGQ, an offline RL method that tries to achieve the best of both worlds, by combining 1-step and chunk-based critics. The authors provide theoretical reasons to justify their methodology, results on OGBench that empirically validate their algorithm, and provide an extensive set of ablation studies.

**Compliance With Llm Reviewing Policy:**

Affirmed.

**Final Justification:**

Given the rebuttal, I will be keeping my score the same.

**Key Questions For Authors:**

- In Eqn. 4, you should acknowledge (and ideally discuss) the loss weight alpha. Would you say that it is equivalent to beta in Eqn. 12, or not?
- Can you provide a better explanation of the difference between DQC and CGQ (ideally to be included in the main paper)? I can acknowledge that there are differences, but it would be appreciated if these could be made clearer.

**Limitations:**

The authors highlighted and discussed negative results in their paper, which was very much appreciated. Other limitations were also fairly mentioned.

**Strengths And Weaknesses:**

Overall, I believe this is a strong paper. The research agenda is presented well, all parts are concise (in a good way), and the authors have both theory (note that I have skimmed this, but have not deeply tried to verify this) and empirical results to justify their method. During reading, I had several questions (e.g., why this vs. using n-step learning), but all of the major questions were answered with their Q&A/ablations. Finally, the appendix has lots of details to aid reproducibility of this work.

It seems important to tune beta; results were presented for one environment, so it is unclear how this changes in other environments. Also, the action-chunked critic may not be the best choice for some environments. However, these are not critical limitations, and the authors did a good job of highlighting these limitations in the main paper.

---

> ### Author Rebuttal · Authors · 2026-03-31
>
> We thank the reviewer for the detailed review and constructive feedback.
>
> &nbsp;
>
> **W1. CGQ vs N-step TD:  I had several questions (e.g., why this vs. using n-step learning)**
>
> Both n-step TD and CGQ aim to reduce bootstrapping error accumulation, but differ fundamentally in how they achieve this.
> N-step TD modifies the TD target directly by replacing the single-step backup with a multi-step return. However, it still relies on the same single-step critic for bootstrapping at the end of the rollout. If the critic's estimates are biased, those errors feed back into the target, perpetuating the very problem n-step TD was meant to solve.
> In contrast, CGQ's action-chunked critic is trained independently from the single-step critic and never directly enters the single-step TD target. It instead serves as a stable external reference that regularizes the single-step critic, providing consistent guidance without reintroducing error accumulation through the bootstrapping chain.
> &nbsp;
>
> &nbsp;
>
> **W2. Sensitivity to β: It seems important to tune beta; results were presented for one environment, so it is unclear how this changes in other environments.**
>
> Thank you for raising this question. We extended the sensitivity analysis for β to one navigation task (antmaze-large) and one non-cube manipulation task (scene). Moreover, tuning β is indeed important: it controls the strength of chunk guidance, and optimal β depends on how reliable that guidance is.
>
> Specifically, we observe the same tendency as cube-double in scene, where stronger regularization (β=0.1, 1) is generally preferred. On the other hand, in antmaze, weaker regularization is preferred, suggesting that when chunk guidance is less reliable, its influence should be reduced.
>
> | Domain        | 0.001     | 0.01      | 0.1       | 1         |
> |-------|------|-------|-------|--------|
> | antmaze-large | 72 ± 2    | 67 ± 3    | 37 ± 3    | 23 ± 5    |
> | scene         | 46 ± 2    | 57 ± 3    | 72 ± 1    | 62 ± 2    |
>
>
>
> &nbsp;
>
> **W3. The action-chunked critic may not be the best choice for some environments**
>
> We agree that the action-chunked critic may not be the best choice for all environments. As discussed in Section 5.2, navigation tasks present a case where the action-chunked critic is unreliable due to the difficulty of learning action-chunked policies in locomotion domains, and the chunk guidance can become harmful in such settings. In contrast, the strong manipulation results demonstrate that when the action-chunked critic is of reasonable quality, CGQ effectively leverages it to improve value estimation.
>
> We acknowledge this as a limitation and improving chunked critic or utilizing other horizon reduction techniques to stabilize value learning as a promising direction.
>
>
>
>
> &nbsp;
>
> **W4. In Eqn. 4, you should acknowledge (and ideally discuss) the loss weight alpha. Would you say that it is equivalent to beta in Eqn. 12, or not?**
>
> Thank you for the question. We’d like to clarify that α in Eq. 4 and β in Eq. 12 serve distinct roles. Specifically, α is the behavioral constraint coefficient in the FQL policy extraction objective, which regularizes the learned policy to remain close to the behavior distribution. In contrast, β in Eq. 12 controls the strength of regularization toward the action-chunked critic. We acknowledge that we did not explicitly discuss α in the main text, which may have caused the confusion, and we will add a clarification in the final draft.
>
>
>
>
> &nbsp;
>
>
> **W5: Can you provide a better explanation of the difference between DQC and CGQ (ideally to be included in the main paper)? I can acknowledge that there are differences, but it would be appreciated if these could be made clearer.**
>
> Thank you for the suggestion. While DQC and CGQ both utilize a large-chunk critic alongside a smaller critic, we'd like to clarify that the motivation and mechanism are fundamentally different.
>
> DQC focuses on improving policy extraction from chunked critics. DQC first trains a large-chunk critic via chunked TD learning, and then distills it into a smaller-chunk critic to ease policy learning. Importantly, the smaller critic is trained solely through distillation and does not use smaller-step TD updates.
>
> In contrast, CGQ begins from the observation that chunked value learning is itself suboptimal due to the use of open-loop action sequences (Eq. 8). Accordingly, CGQ trains the single-step critic directly with standard single-step TD learning, using the chunk critic only as a regularization signal. This preserves the fine-grained value propagation and trajectory stitching benefits of single-step TD learning.
>
> We will include the clarification on this distinction in the Related Works section of the revised paper.

---

> > ### Author Rebuttal · Reviewer_Xo5v · 2026-04-02
> >
> > Thank you for the clarifications. I will be keeping my original recommendation.

---

### Official Review · Reviewer_JpxD · 2026-02-28

**Soundness:** 3
**Presentation:** 4
**Significance:** 4
**Originality:** 3
**Overall Recommendation:** 4
**Confidence:** 4

**Summary:**

This paper proposes Chunk-Guided Q-Learning (CGQ), an RL framework that aims to combine the complementary strengths of single-step TD learning and action-chunked TD learning. Specifically, CGQ regularizes a single-step critic toward an auxiliary chunk-based critic, thereby mitigating bootstrapping error accumulation while preserving fine-grained value estimation.
The proposed method is simple yet conceptually clean. The paper further provides a theoretical analysis showing that, under a bounded curvature assumption of the operator, CGQ achieves a tighter optimality bound than either single-step TD or action-chunked TD learning.

**Compliance With Llm Reviewing Policy:**

Affirmed.

**Final Justification:**

- After rebuttal -
I increased my overall score to 4.
The results on OGBench manipulation tasks would highlight an interesting research question for future study. I wish the results on these two additional manipulation tasks (humanoidmaze-large, antsoccer-arena) were incorporated into Table 1 in the final version.

**Key Questions For Authors:**

Could you provide more insightful explanations or additional empirical results addressing the weaknesses raised above?

**Limitations:**

I believe that addressing the weaknesses would significantly improve the overall quality, clarity of this work.

**Strengths And Weaknesses:**

**Strengths**

S1. (Simple and Conceptually Clean Method)
The proposed method is simple and easy to understand. By introducing an upper-expectile regularization term (eq. 11&12) that guides the single-step critic toward a chunk-based critic. This design is intuitive and practically implementable.

S2. (Theoretical Support)
The paper provides a non-trivial theoretical analysis (in Section 4.2), which formalizes a bias–variance trade-off induced by chunk regularization. Under a bounded curvature assumption, the optimality bound can be tighter than those of pure single-step TD and pure chunk-based learning. The analysis seems to be rigorous, however, the derivations rely on non-trivial assumptions (e.g., bounded curvature), and verifying the proof in full detail seems to require a background in functional analysis. I did not exhaustively check every derivation step for full mathematical rigor.
Nevertheless, the theoretical results provide meaningful intuition: CGQ can interpolate between variance-dominated and bias-dominated terms and identify a finite optimal regularization strength $\beta$.

S3. (Clear Organization and Writing Quality)
The paper is well organized and easy to follow. The overall manuscript is presented in a logical sequence.

**Weaknesses**

W1. (Gap Between Motivation and Practical Objective)
In Figure 2, the proposed method ("CGQ") is illustrated as a weighted sum of single-step and action-chunked value learning (eq.19), as described in Appendix A.5, and it appears a weighted sum rather than chunk-guided learning. However, the actual objective (eq. 11) contains an expectile regularization of the single-step critic toward the chunk-based critic, rather than a simple weighted sum of both learning procedures.
Thus, it is unclear whether the claim in the motivation (the method combines the advantages of both previous value learning approaches) holds under the actual CGQ objective (eq. 12) in practice.
I suggest either revising the description in Figure 2 (e.g., replacing “CGQ” with a clearer phrasing such as “combine both”), or providing experimental results that directly correspond to the exact objective (eq. 12). This would improve consistency between the conceptual motivation and the proposed algorithm.

W2. (Limited Evidence of Improvement Over Single-Step Baseline)
Table 1 shows that CGQ consistently outperforms chunk-based baselines, supporting the claim that it mitigates chunk-induced suboptimality.
However, another claim that CGQ improves upon single-step learning (FQL) is less consistently supported: (i) in navigation tasks, CGQ underperforms FQL, (ii) the improvement margins over FQL are concentrated in "cube-*" tasks, (iii) in "puzzle-3x3-sparse", FQL is already strong. To more convincingly demonstrate improvement over the single-step baseline, I recommend additional experiments:

- Semi-sparse reward setting ("puzzle-3x3")
- More navigation tasks ("humanoidmaze-large", "antsoccer-arena")
- Pixel-based tasks to verify robustness beyond low-dimensional state spaces.

Since CGQ uses a single-step actor and critic, direct comparison against FQL under more diverse tasks would more clearly validate its advantage over single-step TD learning.
Although recent single-step flow-based methods exist (in ICLR 2026), CGQ is a general framework that could add on such methods, therefore, adding further single-step baselines may not be strictly necessary.

W3. (Limited Task Coverage in Ablation Studies)
The ablation studies in Section 5.3 are conducted primarily on cube-double, which is one of the strongest-performing domains for CGQ, as in W2.
To ensure generality, I recommend including additional task category, particularly from navigation domains where chunk critics may be less reliable, as described in Section 5.2.
For example, applying "CGQ-Max" variants to navigation tasks, which could verify the claim that inaccurate chunk-based critics can hurt the performance for the proposed method. This would strengthen the empirical claims regarding when chunk guidance helps or hurts.

W4. (Dealing with distribution shift)
The proposed method is not restricted to offline settings. It appears naturally extendable to offline-to-online settings, as in FQL.
Providing results in an offline-to-online RL setting would strengthen the paper and demonstrate broader applicability.

Overall, this paper presents a well-motivated, theoretically grounded, and conceptually clean approach. However, the paper would benefit from additional empirical results and validation.

---

> ### Author Rebuttal · Authors · 2026-03-31
>
> We thank the reviewer for the thoughtful review, especially on the motivation and objective consistency and the request for broader evidence.
>
> &nbsp;
>
> **W1. Gap Between Motivation and Practical Objective**
>
>
> Thank you for pointing this out. We agree the current labeling in Fig. 2 can be confusing. Fig. 2 and Eq. 19 was intended only as a conceptual illustration of the trade-off between single-step and chunked learning, not as the exact objective optimized by CGQ. We will rename the figure (e.g. “Illustration: convex combination”) and explicitly clarify it in the caption. Our actual objective (Eq. 12) instantiates the same idea more robustly by using the chunked critic only as an auxiliary guidance signal via expectile distillation, while keeping single-step TD as the primary learning objective.
>
> &nbsp;
>
> **W2. Limited Evidence of Improvement Over Single-Step Baseline**
>
> We agree that improvements over FQL are not uniform across all domains. To provide broader evidence, we have added the following experiments beyond the cube-* tasks:
>
> **Semi-sparse reward setting**
> | Task | CGQ | FQL | QC-FQL |
> |-|-|-|-|
> | puzzle-3x3 | **85.3  ± 6**| 32 | 62 |
>
> In the harder semi-sparse reward setting, CGQ substantially improves over FQL, where stable long-horizon propagation is more critical.
>
> &nbsp;
>
> **Additional navigation tasks**
> | Task | CGQ | FQL |
> |-|-|-|
> | antsoccer-arena  | 63.4  ± 2 | 60 |
> | humanoidmaze-large | 0.5  ± 0 | 4 |
>
> CGQ can match FQL on some navigation tasks (63.4 vs. 60 on antsoccer-arena) but can underperform on others (0.5 vs. 4 on humanoidmaze-large), where both methods have low success–consistent with our observation that chunked critics can be unreliable in highly reactive locomotion domains.
>
> &nbsp;
>
> **Pixel-based tasks (single frame visual observations)**
>
> | Task  | CGQ | FQL|
> |-|-|-|
> | visual-cube-single-task1 | **49 ± 12** | 25 ± 7 |
> | visual-cube-double-task1 | **19 ± 7** | 1 ± 7 |
> | visual-scene-play-task1 | **93 ± 1** | 88 ± 4 |
> | visual-puzzle-3x3-task1 | **47 ±  14** | 33 ± 5 |
> | visual-puzzle-4x4-task1 | **21 ± 2**  | 10 ± 3 |
> | Average | **46 ± 2** | 34 ± 2 |
>
> CGQ consistently outperforms FQL on all five visual tasks, suggesting the gains are not limited to low-dimensional states.
>
> &nbsp;
>
> **W3.  Limited Task Coverage in Ablation Studies**
>
> Thank you for the suggestion. We extended the ablations from Sec. 5.3 to include two additional long-horizon tasks: scene and antmaze-large.
>
> **Critic design**
>
> | Domain | CGQ (Ours) | Opposite | Distill | Max |
> |-|-|-|-|-|
> | antmaze-large | **67 ± 3** | 0 ± 0 | 26 ± 5  | 18 ± 1 |
> | scene| **72 ± 1** | 69 ± 1 | 41 ± 2 | 61 ± 1 |
>
> Across both tasks, alternatives underperform CGQ; Opposite is only close on scene where a chunked critic is already strong.
>
> &nbsp;
>
> **Regularization strength $\beta$**
>
> | Domain | 0.001| 0.01| 0.1| 1|
> |-|-|-|-|-|
> | antmaze-large | 72 ± 2  | 67 ± 3 | 37 ± 3 | 23 ± 5 |
> | scene | 46 ± 2 | 57 ± 3 | 72 ± 1 | 62 ± 2 |
>
> This directly supports our “when chunk guidance helps/hurts” claim: navigation prefers smaller $\beta$, while manipulation benefits from moderate $\beta$.
>
> &nbsp;
>
> **Chunk size**
>
> Larger chunks help in manipulation but hurt in navigation, consistent with the difficulty of open-loop chunk learning in reactive locomotion.
>
> | Domain| 5  | 10 | 15 | 20  |
> |-|-|-|-|-|
> | antmaze-large | 65 ± 11 | 67 ±3  | 46 ± 8 | 18 ± 9 |
> | scene | 61 ± 4  | 72 ± 1 | 69 ± 3 | 72 ± 4 |
>
> &nbsp;
>
> **Distillation Expectile $\tau$**
>
> CGQ remains robust to the choice of $\tau$ across both scene and antmaze, consistent with our observation on cube-double.
>
> | Domain  | 0.50  | 0.60 | 0.70  | 0.80 | 0.90 | 0.95 |
> |-|-|-|-|-|-|-|
> | antmaze-large | 70 ± 5  | 66 ± 9  | 67 ± 3  | 62 ± 7 | 56 ± 10 | 59 ± 4 |
> | scene | 68 ± 2  | 67 ± 1  | 69 ± 3 | 69 ± 2 | 70 ± 2 | 72 ± 1 |
>
> &nbsp;
>
> **W4.  Dealing with distribution shift (Offline to Online)**
>
> Thank you for raising this point. CGQ naturally extends to offline-to-online training, as it only modifies critic learning with the guidance signal. We ran offline-to-online RL (following FQL) on one navigation task (humanoidmaze-medium) and two manipulation tasks (cube-double, puzzle-4x4).
> | Task | CGQ | FQL | QC-FQL |
> |-|-|-|-|
> | humanoidmaze-medium-navigate-singletask-v0 |  7 ± 7 -> **34 ± 16** | 12 → 22 | 0 ± 0 -> 0 ± 0 |
> | cube-double-play-singletask-v0 | 40 ± 3 -> **98 ± 2** | 40 → 92  |  43 -> 100 |
> | puzzle-4x4-play-singletask-v0 |17  ± 5 -> **100 ± 0** | 8 → 38 | 0 ± 0 -> 75 ± 50 |
>
> CGQ successfully improves with online fine-tuning across all tasks, and shows notable gains over FQL on puzzle-4x4 (17→100 vs. 8→38), suggesting CGQ's chunk guidance remains beneficial beyond purely offline training.

---

> > ### Author Rebuttal · Reviewer_JpxD · 2026-04-04
> >
> > I thank you for your rebuttal. While most of my concerns have been addressed, one main concern still remains.
> >
> > - It still lacks a sufficiently intuitive and convincing explanation for why CGQ and the Q-chunk-based baselines underperform FQL, a 1-step method, on the OGBench navigation tasks.
> >
> > Despite this remaining concern, I have found the proposed method meaningful. The idea of Q-chunk-guided value learning seems to be a simple yet effective way of combining 1-step and n-step value learning, and the method demonstrates strong performance on the OGBench manipulation tasks.
> >
> > In addition, the rebuttal provided results on two additional manipulation tasks to address the main concern above, which I view positively. I think that these results on OGBench manipulation tasks highlight an interesting research question for future study. It would be helpful if the results on these two additional manipulation tasks were incorporated into Table 1 in the final version.
> >
> > Therefore, I will be increasing my score to 4.

---

> > > ### Author Response · Authors · 2026-04-07
> > >
> > > Thank you for the follow-up and for positive feedback. We address the remaining concern below and will incorporate this explanation in Sec. 5.2 in the revision.
> > >
> > > **Why CGQ and the Q-chunk-based baselines underperform FQL, a 1-step method, on the OGBench navigation tasks?**
> > >
> > >
> > > OGBench navigation (locomotion) tasks require **highly reactive, feedback control**: small state changes (e.g., contact, slippage, turning) immediately change the best action. Action chunking, however, evaluates and optimizes **open-loop action sequences** executed for $h$ steps, which is a poor match for such reactive dynamics. This mismatch makes action-chunked policies difficult to learn and, critically, makes the **chunk critic less reliable** because its value must predict returns under an open-loop commitment that is often suboptimal in locomotion.
> > > As a result, any method that depends on a chunked critic (QC-FQL, DEAS, DQC, and also CGQ) can be harmed in navigation, whereas purely single-step methods, like FQL avoid open-loop commitment and remain robust.
> > >
> > >
> > >
> > > This interpretation is supported by our ablations on navigation: (1) increasing chunk size consistently degrades performance, indicating that longer open-loop commitments are harder in locomotion; and (2) variants that rely more heavily on chunked components (e.g. Opposite) collapses.
> > >
> > >
> > > | Chunk size | 5  | 10 | 15 | 20  |
> > > |-|-|-|-|-|
> > > | antmaze-large | 65 ± 11 | 67 ±3  | 46 ± 8 | 18 ± 9 |
> > >
> > > | Ablated methods | CGQ (Ours) | Opposite | Distill | Max |
> > > |-|-|-|-|-|
> > > | antmaze-large | **67 ± 3** | 0 ± 0 | 26 ± 5  | 18 ± 1 |
> > >
> > >
> > > We will incorporate this explanation and the above evidence to Section 5.2 (navigation results) in the revised version. We also agree that it would be helpful to include the two additional manipulation tasks in Table 1, and we will do so in the final version.
> > >
> > > We would be happy to address any additional concerns you may have, so feel free to let us know.

---

### Official Review · Reviewer_VMuX · 2026-03-02

**Soundness:** 3
**Presentation:** 3
**Significance:** 2
**Originality:** 3
**Overall Recommendation:** 4
**Confidence:** 3

**Summary:**

This paper proposes Chunk-Guided Q-Learning (CGQ), which regularizes a single-step critic toward an auxiliary action-chunked critic, combining the stability of chunk-based TD learning with the optimality of single-step TD learning. The authors provide theoretical analysis showing tighter critic optimality bounds compared to either approach alone. Empirically, CGQ demonstrates strong performance on OGBench manipulation tasks, outperforming both single-step (FQL) and action-chunked (QC-FQL, DEAS, DQC) baselines. However, navigation tasks show mixed results where CGQ underperforms pure single-step methods.

**Compliance With Llm Reviewing Policy:**

Affirmed.

**Final Justification:**

My concerns are addressed. I keep my score of 4.

**Key Questions For Authors:**

Please see the weaknesses to address the main concerns.

**Limitations:**

yes

**Strengths And Weaknesses:**

**Strengths**
1. The paper is well-presented and easy to follow, the motivation is clear to me.
2. The regularization framework blending single-step and chunk-based critics is conceptually elegant and novel.
3. Theoretical analysis on the optimality bounds are provided.

**Weaknesses and Questions**
1. Navigation tasks show CGQ underperforms FQL. The paper acknowledges this but doesn't provide sufficient analysis on when chunk guidance becomes harmful.
2. Does the authors test CGQ in standard D4RL benchmark?
3. How does CGQ's computational overhead compare to baselines?
4. (minor) There is a small blank space at the end of page 8. I suggest filling the space completely. Although this is permitted by the rules, it will affect the first impression when reviewing.

---

> ### Author Rebuttal · Authors · 2026-03-31
>
> We thank the reviewer for the thoughtful and detailed review. We especially appreciate the clarification questions about performances on navigation tasks.
>
> &nbsp;
>
> **W1. Navigation tasks show CGQ underperforms FQL. The paper acknowledges this but doesn't provide sufficient analysis on when chunk guidance becomes harmful.**
>
>
>  Thank you for raising this point. We believe the underperformance of CGQ in navigation tasks is primarily due to the inherent difficulty of action-chunked policy learning in locomotion domains. Locomotion requires highly reactive control, where action-chunked policies are fundamentally limited by their open-loop nature. This leads to an unreliable chunked critic, making the guidance ineffective.
> Notably, as shown in Table 1, all chunk-based methods (QC-FQL, DEAS, DQC) substantially underperform FQL in navigation, suggesting that the difficulty stems from action-chunked policy and critic learning. We acknowledge this as a limitation and identify improving chunk guidance in locomotion domains as a promising direction for future work.
>
> &nbsp;
>
>
> **W2. Does the authors test CGQ in standard D4RL benchmark?**
>
> Thank you for the suggestion. We report the performance of CGQ on the D4RL benchmark:
> | Task                     | CGQ           | FQL |
> |-----|------|-----|
> | antmaze-umaze-v2         | 95 ± 1   | 96  |
> | antmaze-umaze-diverse-v2 | 78 ± 7    | 89  |
> | antmaze-medium-play-v2   | 75 ± 5    | 78  |
> | antmaze-medium-diverse-v2| 75 ± 4    | 71  |
> | antmaze-large-play-v2    | 79 ± 10    | 84  |
> | antmaze-large-diverse-v2 | 81 ± 5    | 83  |
> | pen-expert-v1            | 143 ± 2   | 142 |
> | door-expert-v1           | 104 ± 0   | 104 |
> | hammer-expert-v1         | 122 ± 2   | 125 |
> | relocate-expert-v1       | 108 ± 1   | 107 |
> | Average |95 ± 2 | 98 |
>
> Overall, the results show that CGQ performs comparably to FQL on D4RL. We note that most of D4RL tasks have been saturated, having limited room for improvements.
>
> &nbsp;
>
> **W3. How does CGQ's computational overhead compare to baselines?**
>
> Thank you for raising this point. We report per-step training and inference time of all methods on a single L40S GPU.
>
>
> | Agent | Train (ms/step) | Inference (ms/step) |
> |-------|----------------|-----------------|
> | FQL   | 2.00    | 0.26     |
> | NFQL  | 2.14    | 0.27     |
> | QC-FQL| 2.11    | 0.26     |
> | DEAS | 2.44 | 0.29 |
> | DQC   | 2.97    | 0.66     |
> | CGQ   | 4.66    | 0.36     |
>
> CGQ’s training cost is approximately 2.3x that of FQL, which is expected as CGQ jointly trains both a single-step and an action-chunked critic. We believe this is a reasonable trade-off given the substantial performance gains, outperforming FQL by 33% (38% $\rightarrow$ 71%) on manipulation tasks. We will include this table in the Appendix for the final draft.
>
>
>
>
> &nbsp;
>
>
>
> **W4. (minor) There is a small blank space at the end of page 8. I suggest filling the space completely. Although this is permitted by the rules, it will affect the first impression when reviewing.**
>
> We appreciate the reviewer pointing this out. We will fill the blank space in the final draft.
>
>
> &nbsp;

---

> > ### Author Rebuttal · Reviewer_VMuX · 2026-04-02
> >
> > Thank the authors for responding. I will keep my positive recommendation.

---

### Official Review · Reviewer_wJZE · 2026-03-12

**Soundness:** 2
**Presentation:** 3
**Significance:** 3
**Originality:** 3
**Overall Recommendation:** 3
**Confidence:** 4

**Summary:**

single-step TD suffers from bias propagation problem due to bootstrapping. Action-chunked TD can mitigate this bias propagatation, but may lead to sub-optimal policy due to open-loop action sequence. To solve this trade-off, this paper proposes Chunk-Guide Q-learning. The main idea is to use a chunk-based Q-value to regularize the single-step TD, such as Q-learning, and aims to provide a fine-grained single-step Q-value. Theorems and experiments show the effectiveness of this method in robot control.

**Compliance With Llm Reviewing Policy:**

Affirmed.

**Key Questions For Authors:**

1. I think the action-chunks, i.e., a_t=(a_t,a_t+1,...,a_t+h-1), is very improtant. It decides the credibility of chunked learning singal to guide the single-step TD. How to evaluate the quality of the action-chunks?
2. Based on above problem, there should be a adpative action-chunks choose algorithm.
3. is it possible to test our algorithm in the real robot control?

**Limitations:**

1. I think the action-chunks, i.e., a_t=(a_t,a_t+1,...,a_t+h-1), is very improtant. It decides the credibility of chunked learning singal to guide the single-step TD. How to evaluate the quality of the action-chunks?
2. Based on above problem, there should be a adpative action-chunks choose algorithm.
3. is it possible to test our algorithm in the real robot control?

**Strengths And Weaknesses:**

Strengths:
1. The paper is easy to follow.
2. The idea (use the chunked learning singal to guide the single-step TD) is interesting, and may be useful in robot control. Moveover, it provide the theoretical analysis.
3. The experiments shows the effectiveness of this method in different chunk size and different regularization parameter.

Weaknesses:
1. I think the action-chunks, i.e., a_t=(a_t,a_t+1,...,a_t+h-1), is very improtant. It decides the credibility of chunked learning singal to guide the single-step TD. How to evaluate the quality of the action-chunks?
2. Based on above problem, there should be a adpative action-chunks choose algorithm.
3. is it possible to test our algorithm in the real robot control?

---

> ### Author Rebuttal · Authors · 2026-03-31
>
> We thank the reviewer for the insightful questions, especially regarding the quality of action chunks and their role in guiding single-step TD learning.
>
> &nbsp;
>
> **W1. I think the action-chunks, i.e., a_t=(a_t,a_t+1,...,a_t+h-1), are very improtant. It decides the credibility of chunked learning signal to guide the single-step TD. How to evaluate the quality of the action-chunks?**
>
> Thank you for raising this important point. In CGQ, the action chunks used in our regularization (Eq. 11) are drawn from the offline dataset, with the first action of each chunk aligned with the single-step action used in standard TD learning. CGQ does not explicitly select or filter action chunks.
>
> That said, we agree that the **reliability** of the chunked critic (and thus the usefulness of chunk guidance) depends on the **quality** of available action chunks in the dataset. For example, Section 5.2 shows that chunk guidance can become harmful when the action-chunked critic is unreliable. This supports the reviewer’s concern and clarifies when chunk guidance is expected to help.
>
> In this work, our goal was to investigate how chunked TD learning can be incorporated into single-step TD learning, so we kept the algorithm simple and avoided introducing additional mechanisms beyond standard practice. Filtering or adaptively weighting chunk guidance based on such reliability signals or quality of action chunks is an  interesting direction for future work.
>
>
> &nbsp;
>
> **W2. Based on above problem, there should be an adaptive action-chunks choose algorithm.**
>
> We agree that adaptive action-chunk selection is a promising direction. While our goal in this paper was to keep the algorithm simple and close to standard practice, the above reliability signal naturally suggests adaptive extensions, e.g., down-weighting the regularizer when the chunk critic is uncertain. As a first step toward this, our $\beta$-sweep already shows that reducing $\beta$ mitigates the negative effect of chunk guidance in locomotion tasks, which is consistent with an adaptive weighting strategy.
>
> | Domain        | 0.001     | 0.01      | 0.1       | 1         |
> |------|-------|------|--------|------|
> | antmaze-large | 72 ± 2    | 67 ± 3    | 37 ± 3    | 23 ± 5    |
> | scene         | 46 ± 2    | 57 ± 3    | 72 ± 1    | 62 ± 2    |
> | cube-double | 32 ± 5 | 37 ± 2 | 58 ± 3 | 64 ± 4 |
>
> &nbsp;
>
> **W3. Is it possible to test our algorithm in the real robot control?**
>
>
>
> We did not evaluate CGQ on real robot hardware in this submission, and we acknowledge this limitation. We evaluated CGQ on OGBench across manipulation and navigation domains (45 tasks total), which includes long-horizon robot manipulation and locomotion benchmarks. Since CGQ only adds an auxiliary chunk critic and a lightweight regularization term on top of standard offline RL training, it is compatible with existing real-robot pipelines, and we plan to pursue real-world validation in future work.
>
> &nbsp;
>
> We would like to thank you again for raising important points about our work. Please let us know if you have any additional concerns or questions.

---

> > ### Author Rebuttal · Reviewer_wJZE · 2026-04-01
> >
> > Thanks for your discussion.
> > Section 5.2 shows that chunk guidance can become harmful when the action-chunked critic is unreliable.
> > This confirms my concern.
> > Then, how to evaluate the quality of the action-chunks?
> > without a clear mechanism to distinguish helpful action-chunks or harmful action-chunks is unaccepted.
> > Then, I keep my score unchange.

---

> > > ### Author Response · Authors · 2026-04-07
> > >
> > > Thank you for the follow-up and for the valuable suggestion. We agree, and we investigated two explicit mechanisms to identify helpful vs. harmful action chunks, and used them to filter out potentially harmful chunks in CGQ:
> > >
> > > **(1) Q-value disagreement (Q-value filtering).** We measure the relative difference between the chunked value and the corresponding single-stepe value: $$\\frac{Q_c(s, a_{t:t+h}) - Q(s, a_t)}{|Q(s, a_t)|}$$ We **filter out action chunks whose ratio exceeds a threshold, treating large disagreements as unreliable for guidance**. We sweep the threshold over [0.05, 0.1, 0.2, 0.3, 0.5, 0.6] for each task.
> > >
> > > **(2) Chunked TD-error (Q-loss filtering).** We compute the per-chunk TD error: $$ (Q_c(s_t, \mathbf a_t)-\sum_{i=0}^{h-1}\gamma^i r_{t+i}-\gamma^h \bar Q_c(s_{t+h},\mathbf{a}_{t+h}))^2 $$ We **filter out action chunks above the $\tau$-th quantile of this loss**, treating poorly fitted chunks as unreliable for guidance. We sweep $\tau$ over [0.2, 0.4, 0.6, 0.8, 0.9, 0.99] for each task.
> > > | Domain | CGQ | Q-value filtering | Q-loss filtering |
> > > |-|-|-|-|
> > > | antmaze-large (5 tasks)| 67 ± 3 |71 ± 6 |  **79 ± 5** |
> > > | scene (5 tasks) | **72 ± 1** | **71 ± 1** | **72 ± 1** |
> > >
> > >
> > > Both filtering methods substantially improve performance on navigation tasks, where harmful action chunks are prevalent (consistent with our observation in Sec 5.2). This indicates that the evaluation mechanisms successfully identify and suppress unreliable chunks during guidance. On manipulation (scene), filtering has little effect, consistent with our observation that harmful chunks rarely appear in these domains. We will include this analysis in the revised paper.
> > >
> > > We would like to thank you again for the important suggestion. Please let us know if you have any additional concerns or questions.

---

### Decision · Program_Chairs · 2026-04-30

**Decision:**

Accept (regular)

**Comment:**

This paper proposes Chunk-Guided Q-Learning (CGQ), which integrates single-step TD learning with chunk-based critics via a regularization mechanism to balance fine-grained value propagation and long-horizon stability in offline RL. The paper is clearly written, conceptually clean, and supported by both theoretical analysis and empirical results. Reviewers generally agree that the approach is novel, technically sound, and practically relevant, particularly for long-horizon manipulation tasks.

Strengths include a principled framework that bridges single-step and chunk-based learning, theoretical guarantees on improved optimality bounds, and strong empirical performance over chunk-based baselines. The rebuttal further improves the paper by adding experiments (e.g., semi-sparse rewards, pixel-based tasks, offline-to-online settings) and clarifying differences from related methods.

However, some concerns remain. The empirical advantage over strong single-step baselines (e.g., FQL) is inconsistent, especially in navigation tasks where CGQ can underperform. While the authors provide a reasonable explanation based on the mismatch between action chunking and reactive control, this limits generality. Additionally, the reliability of the chunk-based critic remains a key issue. One reviewer notes that the lack of a principled mechanism for evaluating or selecting high-quality action chunks weakens the method, though rebuttal experiments suggest promising directions.

Overall, the paper presents a solid and meaningful contribution. Despite some limitations in robustness and generality, the majority of reviewers are positive after rebuttal.